# Data-Dependent Regret and Polyak Corrections for Constrained Online Convex Optimization

**Wentao Zhang**                                        *zhang-wt24@mails.tsinghua.edu.cn*
*Tsinghua Shenzhen International Graduate School*
*Tsinghua University*

## Abstract

In constrained online convex optimization, the learner must minimize regret against adversarially chosen convex costs while satisfying a convex constraint at every round, a requirement that arises naturally in safety-critical domains such as power systems, autonomous control, and clinical decision-making. A natural and computationally efficient approach augments online gradient descent with a Polyak feasibility step: a closed-form half-space projection requiring only one constraint evaluation and one subgradient per round. This approach is known to achieve $O(\sqrt{T})$ regret with per-round feasibility, yet we show that its existing analysis admits a tighter, data-dependent form, by retaining two quantities that the standard worst-case argument does not need to track. Specifically, replacing the worst-case gradient envelope $G_f^2 T$ with the observed accumulation $\mathcal{G}_T = \sum_t \|\nabla f_t(x_t)\|^2$ yields a data-dependent bound without any algorithmic modification. Furthermore, we identify the Polyak correction $\mathcal{P}_T \geq 0$, which captures the cumulative squared displacement of the feasibility projection and enters the regret bound with a strictly negative sign; this term equals the slack in the strong (rather than weak) Pythagorean inequality and, while implicit in standard convex analysis, has not been instantiated as a measurable Polyak quantity in prior OCO regret bounds. The total improvement $\Delta_T = \frac{\eta}{2}(G_f^2 T - \mathcal{G}_T) + \frac{1}{2\eta}\mathcal{P}_T$ is provably non-negative and decomposes into two independent, complementary sources that vanish only in a degenerate corner case. Building on these analytical insights, we propose AdaOGD-PFS, an adaptive-step-size variant that achieves $O(\sqrt{\mathcal{G}_T})$ regret, potentially much smaller than $O(G_f\sqrt{T})$, while preserving per-round constraint satisfaction. Experiments on ball-constrained and halfspace-constrained instances confirm bound improvements of 38–43%, with both data-dependent gradients and Polyak corrections contributing meaningfully.

## 1 Introduction

Online convex optimization (OCO) provides a principled framework for sequential decision-making under uncertainty, with applications spanning portfolio selection (Cover, 1991), network routing (Awerbuch & Kleinberg, 2004), and real-time resource allocation (Mahdavi et al., 2012). In many safety-critical deployments (power grid management, autonomous navigation, clinical dosing), decisions must satisfy hard constraints at every round, since even transient violations may incur catastrophic consequences. This practical necessity has driven a surge of interest in constrained OCO, where the learner must simultaneously minimize cumulative regret $\text{Reg}_T = \sum_{t=1}^T f_t(x_t) - \min_{x \in \mathcal{X}} \sum_{t=1}^T f_t(x)$ against an adversarially chosen sequence of convex cost functions $f_1, \ldots, f_T$ while respecting a convex constraint $g(x) \leq 0$ with limited feedback about $g$. Among the algorithms designed for this setting, the combination of Online Gradient Descent (OGD) with *Polyak feasibility steps*, which project onto a first-order approximation of the feasible set using only the constraint value $g(x_t)$ and a single subgradient $s_t \in \partial g(x_t)$, has emerged as a particularly attractive approach due to its computational simplicity and strong feasibility guarantees (Mahdavi et al., 2012; Polyak, 1969).

The study of constrained OCO has progressed through several stages of increasingly refined algorithms. The classical approach of Zinkevich (2003) achieves $O(\sqrt{T})$ regret via projected OGD, but requires computing the full projection $\Pi_{\mathcal{X}}(y_t)$ at each round, a step that is computationally prohibitive when the feasible set

Table 1: Comparison of regret bounds for constrained OCO with Polyak feasibility steps. All three use the same assumptions (1–4) and the same constraint feedback ($g(x_t)$ and one subgradient per round). Here $\mathcal{G}_T = \sum_t \|\nabla f_t(x_t)\|^2 \le G_f^2 T$, $\mathcal{P}_T = \sum_t \delta_t \ge 0$, and $\Delta_T = \frac{\eta}{2}(G_f^2 T - \mathcal{G}_T) + \frac{1}{2\eta}\mathcal{P}_T \ge 0$ (Corollary 4).

| | Hutchinson & Alizadeh (2025) | Theorem 1 | Theorem 2 |
|---|---|---|---|
| Algorithm | OGD-PFS (fixed $\eta$) | **Same** | **AdaOGD-PFS** (adaptive $\eta_t$) |
| Assumptions | Assumptions 1–4 | **Same** | **Same** |
| Constraint feedback | $g(x_t)$, $\partial g(x_t)$ | **Same** | **Same** |
| Knowledge of $G_f$ | Required | Required | **Not required** |
| Regret bound | $\frac{2R^2}{\eta} + \frac{\eta}{2}G_f^2 T + \frac{G_f\rho}{\sigma}T$ | $\frac{2R^2}{\eta} + \frac{\eta}{2}\mathcal{G}_T - \frac{1}{2\eta}\mathcal{P}_T + \frac{G_f\rho}{\sigma}T$ | $2R\sqrt{2(\mathcal{G}_T + \epsilon_0)} + \frac{cG_f^2}{2\sqrt{\epsilon_0}} - \sum_t \frac{\delta_t}{2\eta_t} + \frac{G_f\rho}{\sigma}T$ |
| Feasibility | $g(x_t) \le 0$, $\forall t$ | **Same** | **Same** |
| Worst-case | $O(G_f\sqrt{T})$ | $O(G_f\sqrt{T})$ | $O(G_f\sqrt{T})$ |
| Instance-dep. | — | $O(G_f\sqrt{T}) - \Delta_T$ | $O(\sqrt{\mathcal{G}_T})$ (**best**) |

$\mathcal{X} = \{x : g(x) \le 0\}$ is defined by a general convex function. To circumvent this bottleneck, Mahdavi et al. (2012) replaced exact projection with a single subgradient query per round, attaining $O(\sqrt{T})$ regret with $O(T^{3/4})$ cumulative constraint violation. Subsequent work reduced the violation to $O(\sqrt{T})$ (Jenatton et al., 2016; Yu & Neely, 2020) and further to $O(1)$ using budget-management techniques (Liakopoulos et al., 2019; Neely & Yu, 2017), but these methods generally cannot guarantee per-round feasibility $g(x_t) \le 0$ for all $t$. More recently, OGD was augmented with a Polyak feasibility step (Hutchinson & Alizadeh, 2025), a closed-form half-space projection that exploits the linearization of $g$ at the current iterate, establishing $O(\sqrt{T})$ regret with per-round feasibility when a strictly feasible starting point is known. Despite this algorithmic progress, we observe that the standard regret analysis of the Polyak feasibility step admits a tighter, data-dependent form that has not been instantiated in prior work. Specifically, the standard regret proof applies two coarse relaxations: it upper-bounds each per-round gradient norm $\|\nabla f_t(x_t)\|^2$ by its worst-case value $G_f^2$, replacing the data-dependent sum $\mathcal{G}_T := \sum_{t=1}^{T}\|\nabla f_t(x_t)\|^2$ with the uniform bound $G_f^2 T$; and it invokes the standard non-expansiveness of projection, $\|x_{t+1} - x^\star\|^2 \le \|y_t - x^\star\|^2$, which suffices for the worst-case $O(\sqrt{T})$ rate but does not retain the squared distance $\delta_t = \|y_t - \Pi_{H_t}(y_t)\|^2$ by which the Polyak step moves the intermediate iterate back toward feasibility. The cumulative effect across $T$ rounds is a provably non-negative quantity $\mathcal{P}_T := \sum_{t=1}^{T} \delta_t \ge 0$ that does not appear in existing regret bounds.

In this paper, we provide a *refined regret analysis* of the same OGD + Polyak feasibility step algorithm, using the same assumptions and the same constraint feedback model. Our observation is that the strong (rather than weak) form of the Pythagorean projection inequality (Bauschke & Combettes, 2017, Cor. 4.10) can be used in place of the standard non-expansiveness step to retain the Polyak correction $\delta_t$, and that the per-round gradient norms need not be relaxed to their supremum. Concretely, we prove the following refined regret bound (Theorem 1):

$$\text{Reg}_T \;\le\; \frac{2R^2}{\eta} + \frac{\eta}{2}\underbrace{\sum_{t=1}^{T}\|\nabla f_t(x_t)\|^2}_{\mathcal{G}_T \le G_f^2 T} - \frac{1}{2\eta}\underbrace{\sum_{t=1}^{T}\delta_t}_{\mathcal{P}_T \ge 0} + \frac{G_f\rho}{\sigma}T, \tag{1}$$

which improves upon the prior bound $\frac{2R^2}{\eta} + \frac{\eta}{2}G_f^2 T + \frac{G_f\rho}{\sigma}T$ in two independent and complementary ways. Table 1 provides a systematic comparison between our analysis and prior work.

The two sources of improvement are qualitatively distinct. The gradient refinement ($\mathcal{G}_T$ replacing $G_f^2 T$) captures the fact that the worst-case gradient norm $G_f$, defined as the supremum over the entire bounding ball $R\mathcal{B}$, is typically achieved only by a small fraction of rounds. In our experiments, the ratio $\mathcal{G}_T/(G_f^2 T)$ ranges from 0.27 to 0.31 across problem instances, yielding a 34–37% tightening of the bound. The Polyak correction ($-\frac{1}{2\eta}\mathcal{P}_T$) is the cumulative slack in the strong (rather than weak) projection inequality (Bauschke & Combettes, 2017, Cor. 4.10); while this slack is implicit in classical convex analysis, to our knowledge it has not previously been instantiated as a measurable Polyak-projection quantity in OCO regret bounds. It is

strictly positive whenever the gradient step pushes the iterate outside the linearized constraint, contributing an additional 1–8% improvement. Crucially, our refined bound is never worse than the prior bound (since the improvement $\Delta_T \geq 0$ by construction), and all feasibility guarantees are preserved without modification.

Our main contributions are summarized as follows.

- **Tighter data-dependent regret bound (Theorem 1).** We prove that the standard OGD with a Polyak feasibility step, without any algorithmic modification, admits the refined bound equation 1 by retaining two quantities that prior worst-case proofs do not track. Specifically, we replace $G_f^2 T$ with the observed gradient accumulation $\mathcal{G}_T \leq G_f^2 T$ and identify the Polyak correction $\mathcal{P}_T \geq 0$ that enters with a negative sign and is, to our knowledge, the first instantiation of this slack as a measurable Polyak quantity in OCO regret bounds. The resulting bound is uniformly no worse and strictly tighter in all non-degenerate cases.

- **Adaptive algorithm with data-dependent rate (Theorem 2).** We propose AdaOGD-PFS, which replaces the fixed step size with an adaptive $\eta_t = c / \sqrt{\epsilon_0 + \sum_{i<t} \|\nabla f_i(x_i)\|^2}$, achieving $O(\sqrt{\mathcal{G}_T})$ regret (potentially much smaller than $O(G_f \sqrt{T})$) while preserving per-round feasibility and requiring no knowledge of $G_f$.

- **Empirical validation.** Experiments on ball-constrained and halfspace-constrained instances confirm bound improvements of 38–43% over the prior, with both gradient refinement and Polyak corrections contributing meaningfully.

## 2 Related Work

We situate our contributions within three lines of research: constrained online convex optimization, projection-free and Polyak-type feasibility methods, and data-dependent regret analysis.

**Constrained online convex optimization.** The unconstrained OCO framework, formalized by Zinkevich (2003) and surveyed comprehensively by Hazan (2016) and Shalev-Shwartz (2011), achieves $O(\sqrt{T})$ regret via projected online gradient descent (OGD) when the feasible set admits an efficient projection oracle. Extending this framework to functional constraints $g(x) \leq 0$, where the projection onto $\mathcal{X} = \{x : g(x) \leq 0\}$ may be intractable, has been an active research direction over the past decade.

Mahdavi et al. (2012) initiated this line by proposing an algorithm that queries only a single subgradient of $g$ per round, achieving $O(\sqrt{T})$ regret with $O(T^{3/4})$ cumulative constraint violation. Jenatton et al. (2016) improved the violation to $O(\sqrt{T})$ using adaptive constraint-weighting schemes. A parallel line of work pursued *long-term* constraint satisfaction, where the goal is to bound $\sum_{t=1}^{T} g(x_t)$ rather than enforce $g(x_t) \leq 0$ at each round. Neely & Yu (2017) and Yu & Neely (2020) achieved $O(\sqrt{T})$ regret with $O(1)$ cumulative violation via drift-plus-penalty and virtual-queue techniques rooted in Lyapunov optimization. Liakopoulos et al. (2019) proposed a cautious variant that maintains $O(\sqrt{T})$ regret while keeping the cumulative violation strictly bounded. More recently, Yi et al. (2023) extended constrained OCO to distributed multi-agent settings, Guo et al. (2022) studied time-varying constraints where $g$ itself changes across rounds, and Zhang (2026b) considered the multi-constraint setting with adversarial constraints.

A key limitation shared by these methods is the gap between *cumulative* and *per-round* feasibility: while cumulative violation $\sum_t [g(x_t)]_+$ can be made sublinear or even $O(1)$, ensuring $g(x_t) \leq 0$ at *every* round $t$ requires fundamentally different algorithmic techniques. The Polyak feasibility step was recently introduced into OGD (Hutchinson & Alizadeh, 2025), achieving $O(\sqrt{T})$ regret with per-round feasibility under a known strictly feasible starting point. Our work does not propose a new algorithm for constrained OCO; instead, we demonstrate that the existing regret analysis can be significantly tightened, by up to 43% in our experiments, without modifying the algorithm or its assumptions.

**Projection-free methods and Polyak-type feasibility steps.** When the feasible set $\mathcal{X}$ is complex, computing the Euclidean projection $\Pi_{\mathcal{X}}(\cdot)$ can be as expensive as solving the original optimization problem.

This has motivated a rich body of work on *projection-free* online optimization. The Frank–Wolfe (conditional gradient) method (Frank & Wolfe, 1956; Hazan & Kale, 2012) replaces projection with a linear minimization oracle over $\mathcal{X}$, achieving $O(T^{3/4})$ regret in the general case. Garber & Hazan (2016) improved this to $O(\sqrt{T})$ for polyhedral sets, and Chen et al. (2019) extended the approach to bandit feedback. While these methods avoid full projection, they still require a linear optimization oracle that may be non-trivial for general convex constraints.

A different approach, which is the focus of this paper, replaces the exact projection with a *Polyak-type step* (Polyak, 1969). Originally introduced for convex feasibility problems, the Polyak step projects onto the half-space $H_t = \{x : g(x_t) + s_t^\top (x - x_t) \leq 0\}$ defined by the first-order approximation of $g$ at the current point. This projection has a closed-form solution requiring only $g(x_t)$ and one subgradient $s_t \in \partial g(x_t)$, making it substantially cheaper than full projection. Mahdavi et al. (2012) first used this idea in the OCO context, and subsequent work (Hutchinson & Alizadeh, 2025) introduced the constraint shrinkage parameter $\rho$ to achieve per-round feasibility via the Polyak step applied to the tightened half-space $H_t^\rho = \{x : g(x_t) + s_t^\top (x - x_t) + \rho \leq 0\}$. Our contribution is orthogonal to algorithmic design: we analyze the *same* Polyak feasibility step but extract a tighter regret bound by retaining the squared displacement $\delta_t = \|y_t - \Pi_{H_t}(y_t)\|^2$ that is not tracked by the worst-case argument of prior analyses. This quantity $\delta_t$ is intrinsic to the Polyak step geometry and does not arise in standard projection-based or Frank–Wolfe-based analyses.

**Data-dependent and adaptive regret bounds.** The idea of replacing worst-case constants with data-dependent quantities in regret bounds has a long history in online learning. The AdaGrad algorithm of Duchi et al. (2011) achieves regret bounds that scale with $\sum_t \|\nabla f_t(x_t)\|^2$ rather than $G_f^2 T$, adapting the step size to the observed gradient magnitudes. McMahan (2017) unified several adaptive methods under the follow-the-regularized-leader framework, and Orabona & Pál (2018) developed scale-free algorithms whose bounds automatically adapt to the gradient scale without prior knowledge of $G_f$. In a related vein, Cutkosky & Orabona (2018) established parameter-free regret bounds in Banach spaces, and Steinhardt & Liang (2014) studied the gap between worst-case and adaptive regret in the experts setting. Recent OCO work has also pursued complementary forms of adaptivity, including dynamic regret with untrusted decision predictions (Zhang, 2026a) and noise-adaptive high-probability regret bounds (Zhang et al., 2026).

For unconstrained OCO, data-dependent bounds of the form $O(\sqrt{\sum_t \|\nabla f_t(x_t)\|^2})$ are by now standard and can be achieved algorithmically via adaptive step sizes. In the constrained setting, however, data-dependent bounds have received comparatively little attention. The regret analyses in prior work (Mahdavi et al., 2012; Yu & Neely, 2020; Hutchinson & Alizadeh, 2025) all employ the uniform bound $\|\nabla f_t(x_t)\| \leq G_f$ in the final step of their proofs, collapsing the data-dependent sum $\mathcal{G}_T = \sum_t \|\nabla f_t(x_t)\|^2$ to $G_f^2 T$.

Our work bridges this gap by showing that the data-dependent quantity $\mathcal{G}_T$ can be preserved in the analysis of constrained OGD with a Polyak step without any algorithmic modification. The standard fixed-step-size OGD already enjoys this tighter bound, indicating that the prior analysis can be sharpened. The Polyak correction $\mathcal{P}_T$ that we identify is a data-dependent quantity specific to the constrained setting that equals the slack term in the strong projection inequality (Bauschke & Combettes, 2017, Cor. 4.10) and is the constrained-OCO analogue of the projection slack tracked, in different forms, by self-confident and AdaGrad-style analyses (Auer et al., 2002; Duchi et al., 2011); we are not aware of an OCO regret bound that has previously instantiated it as a measurable Polyak-projection quantity. For the fixed-step-size algorithm, our refined bound serves as a complement to AdaGrad-style adaptive methods by providing a tighter post-hoc characterization. We further bridge the gap algorithmically by introducing AdaOGD-PFS (Section 4.1), which transports the self-confident and AdaGrad step-size design (Auer et al., 2002; Duchi et al., 2011) into the constrained Polyak-feasibility setting and combines it with the Polyak correction to obtain an $O(\sqrt{\mathcal{G}_T})$ regret bound.

To position this contribution against the self-confident and AdaGrad-style line of work, we note that the $O(\sqrt{\mathcal{G}_T})$ scaling we obtain for AdaOGD-PFS is, in the unconstrained setting, the classical self-confident rate of Auer et al. (2002) and the AdaGrad rate of Duchi et al. (2011); the design of $\eta_t = c/\sqrt{S_t}$ is theirs and we make no claim of a new data-dependent tuning principle. Our contribution is to transport this design into the constrained OGD-with-Polyak-feasibility-step setting while preserving per-round constraint satisfaction (Hutchinson & Alizadeh, 2025), which requires controlling the AdaGrad telescoping in the presence of the

Table 2: Summary of notation. Symbols above the mid-rule are problem primitives; those below are derived quantities introduced in this work.

| Symbol | Definition |
|---|---|
| $d$ | Dimension of the decision space |
| $T$ | Total number of rounds (time horizon) |
| $f_t : \mathbb{R}^d \to \mathbb{R}$ | Convex cost function revealed at round $t$ |
| $g : \mathbb{R}^d \to \mathbb{R}$ | Fixed convex constraint function |
| $\mathcal{X} := \{x : g(x) \leq 0\}$ | Feasible set |
| $\mathcal{X}_\rho := \{x : g(x) \leq -\rho\}$ | Shrunk feasible set ($\rho \geq 0$) |
| $x_t \in \mathbb{R}^d$ | Decision played at round $t$ |
| $y_t \in \mathbb{R}^d$ | Intermediate iterate after gradient step: $y_t = x_t - \eta \nabla f_t(x_t)$ |
| $g_t := g(x_t)$ | Constraint value at round $t$ |
| $s_t \in \partial g(x_t)$ | Constraint subgradient at round $t$ |
| $\eta > 0$ | Learning rate (step size) |
| $\rho \geq 0$ | Constraint shrinkage parameter |
| $R, G_f, G_g, \sigma, \epsilon$ | Problem constants (see Assumptions 1–4) |
| $\gamma := 1 - \sigma^2/G_g^2$ | Contraction rate of the Polyak step |
| $\xi := 1 - \sqrt{\gamma}$ | Feasibility convergence rate |
| $H_t$ | Separating half-space: $\{x : g_t + s_t^\top(x - x_t) + \rho \leq 0\}$ |
| $x^\star$ | Offline optimum: $\arg\min_{x \in \mathcal{X}} \sum_{t=1}^T f_t(x)$ |
| $\delta_t := \|y_t - \Pi_{H_t}(y_t)\|^2$ | Polyak correction at round $t$ |
| $\mathcal{P}_T := \sum_{t=1}^T \delta_t$ | Cumulative Polyak correction |
| $\mathcal{G}_T := \sum_{t=1}^T \|\nabla f_t(x_t)\|^2$ | Data-dependent gradient accumulation |

Polyak feasibility step (Theorem 2) and showing that the feasibility recursion in Lemma 4 can be unrolled with the time-varying $\eta_t$ to retain the per-round guarantee. The novel element of the bound, beyond porting the AdaGrad rate, is the inclusion of the Polyak correction $-\sum_t \delta_t/(2\eta_t)$ with a time-varying coefficient that gives later corrections heavier weight than the fixed-step analysis.

## 3   Problem Setup

We consider the constrained online convex optimization (OCO) problem studied in prior work (Mahdavi et al., 2012; Hutchinson & Alizadeh, 2025). A learner interacts with an adversary over $T$ rounds. At each round $t \in [T] := \{1, 2, \ldots, T\}$, the learner selects a decision $x_t \in \mathbb{R}^d$ and the adversary reveals a convex cost function $f_t : \mathbb{R}^d \to \mathbb{R}$. The learner's decisions are subject to a fixed convex constraint $g(x) \leq 0$, about which only first-order information is available. This section defines the notation, the protocol, the standing assumptions, and the performance metrics. The algorithms under study (OGD-PFS and AdaOGD-PFS) are presented in Section 4.

### 3.1   Notation

We write $\|\cdot\|$ for the Euclidean ($\ell_2$) norm on $\mathbb{R}^d$. For a scalar $a \in \mathbb{R}$, we denote $[a]_+ := \max(a, 0)$. The closed Euclidean ball of radius $R$ centered at the origin is $R\mathcal{B} := \{x \in \mathbb{R}^d : \|x\| \leq R\}$. For a closed convex set $S \subseteq \mathbb{R}^d$, we write $\Pi_S(v) := \arg\min_{u \in S}\|v - u\|$ for the Euclidean projection of $v$ onto $S$, and $\text{dist}(v, S) := \min_{u \in S}\|v - u\|$ for the distance from $v$ to $S$. Given a convex function $h : \mathbb{R}^d \to \mathbb{R}$, we write $\partial h(x)$ for its subdifferential at $x$. Table 2 summarizes the principal symbols used throughout the paper.

### 3.2   Protocol

The interaction between the learner and the adversary proceeds as follows. The learner selects an initial point $x_1 \in R\mathcal{B}$. Then, for each round $t = 1, 2, \ldots, T$, the learner commits to the decision $x_t$; the adversary reveals the convex cost function $f_t$ and the learner observes $f_t(x_t)$ and $\nabla f_t(x_t)$; the learner queries the constraint

oracle, receiving $g_t = g(x_t)$ and one subgradient $s_t \in \partial g(x_t)$; and finally the learner updates $x_t \mapsto x_{t+1}$ using $\nabla f_t(x_t)$, $g_t$, and $s_t$.

Two aspects of this protocol merit emphasis. First, the learner receives only *first-order* constraint feedback, namely the function value $g_t$ and a single subgradient $s_t$, rather than access to a projection oracle for $\mathcal{X}$. Second, the cost functions $f_t$ are chosen by an *oblivious* adversary: the entire sequence $(f_1, \ldots, f_T)$ is fixed before the interaction begins, though the learner does not know it in advance.

### 3.3  Assumptions

The following four assumptions are standard in constrained OCO (Mahdavi et al., 2012; Hutchinson & Alizadeh, 2025); we neither strengthen nor weaken them.

**Assumption 1** (Bounded action set). *There exists $R > 0$ such that $\mathcal{X} \subseteq R\mathcal{B}$.*

This is standard in OCO and ensures that the diameter of the feasible set is at most $2R$. It is satisfied whenever $\mathcal{X}$ is compact, which holds in virtually all applications of interest.

**Assumption 2** (Bounded cost gradients). *There exists $G_f > 0$ such that $\|\nabla f_t(x)\| \leq G_f$ for all $x \in R\mathcal{B}$ and all $t \in [T]$.*

This is the standard Lipschitz assumption on the cost functions. It implies $|f_t(u) - f_t(v)| \leq G_f \|u - v\|$ for all $u, v \in R\mathcal{B}$. The constant $G_f$ serves as a worst-case envelope; a central theme of our analysis is that the actual observed quantity $\mathcal{G}_T = \sum_t \|\nabla f_t(x_t)\|^2$ is typically much smaller than $G_f^2 T$.

**Assumption 3** (Bounded constraint subgradients). *There exists $G_g > 0$ such that $\|s\| \leq G_g$ for all $x \in R\mathcal{B}$ and all $s \in \partial g(x)$.*

Together with Assumption 1, this implies that $g$ is $G_g$-Lipschitz on $R\mathcal{B}$. The upper bound $G_g$ governs the contraction rate of the Polyak step via the ratio $\sigma/G_g$.

**Assumption 4** (Constraint boundary subgradient lower bound). *There exist $\sigma > 0$ and $\epsilon > 0$ such that the level set $\mathcal{X}' := \{x \in \mathbb{R}^d : g(x) = -\epsilon\}$ is nonempty, and $\|s\| \geq \sigma$ for all $x \in \mathcal{X}'$ and all $s \in \partial g(x)$.*

This assumption ensures that the constraint function $g$ has a non-vanishing gradient near the boundary of $\mathcal{X}$, which is necessary for the Polyak step to achieve geometric contraction toward feasibility. Assumption 4 is automatically satisfied whenever Assumption 1 holds and the Slater condition is met, i.e., there exists $y$ with $g(y) < 0$. The ratio $\gamma := 1 - \sigma^2/G_g^2 \in [0, 1)$ measures the "condition number" of the constraint: when $\sigma \approx G_g$, the Polyak step contracts rapidly ($\gamma \approx 0$); when $\sigma \ll G_g$, contraction is slower ($\gamma \approx 1$). We also define $\xi := 1 - \sqrt{\gamma} > 0$, which governs the rate at which the feasibility distance decays (see Theorem 1 and Lemma 5).

**Remark 1.** *We emphasize that no assumption beyond Assumptions 1–4 is introduced in this paper. The comparability of our results with those of prior work rests entirely on this fact: any improvement in the regret bound is due to tighter analysis, not stronger assumptions.*

### 3.4  Performance Metrics

We evaluate the learner's performance along two axes.

**Regret.**  The cumulative regret measures the excess cost incurred by the learner relative to the best fixed decision in hindsight:

$$\mathrm{Reg}_T := \sum_{t=1}^{T} f_t(x_t) - \min_{x \in \mathcal{X}} \sum_{t=1}^{T} f_t(x). \tag{2}$$

A sublinear regret guarantee $\mathrm{Reg}_T = O(\sqrt{T})$ implies that the learner's average per-round cost converges to that of the offline optimum as $T \to \infty$.

**Feasibility.** We consider two notions of constraint satisfaction. Per-round feasibility requires $g(x_t) \leq 0$ for all $t \in [T]$ and is the strongest guarantee, needed in safety-critical applications. Cumulative feasibility requires $\sum_{t=1}^{T}[g(x_t)]_+ \leq V(T)$ for some sublinear function $V$ and is a weaker but more commonly studied notion. Our analysis preserves the per-round feasibility guarantee without modification.

### 3.5 Key Quantities Introduced in This Work

Our refined analysis is built around two quantities that are naturally present in the algorithm's iterates but have been overlooked in prior regret bounds.

**Definition 1** (Polyak correction). *For each round $t \in [T]$, the* Polyak correction *is*

$$\delta_t \;:=\; \|y_t - \Pi_{H_t}(y_t)\|^2 \;=\; \frac{[g_t + s_t^\top(y_t - x_t) + \rho]_+^2}{\|s_t\|^2}, \tag{3}$$

*which equals zero when $y_t \in H_t$ and is strictly positive otherwise. The* cumulative Polyak correction *is* $\mathcal{P}_T := \sum_{t=1}^{T}\delta_t \geq 0$.[1]

Geometrically, $\delta_t$ is the squared distance by which the Polyak step moves $y_t$ toward the half-space $H_t$. It is positive precisely in rounds where the gradient step pushes the iterate outside the linearized constraint, i.e., the rounds where the Polyak step does non-trivial "corrective work." The standard non-expansiveness inequality $\|x_{t+1} - x^\star\|^2 \leq \|y_t - x^\star\|^2$ already suffices for the worst-case $O(\sqrt{T})$ rate, so prior analyses do not track $\delta_t$; the strong (rather than weak) Pythagorean inequality (Bauschke & Combettes, 2017, Cor. 4.10) in fact gives $\|x_{t+1} - x^\star\|^2 \leq \|y_t - x^\star\|^2 - \delta_t$, and summing over $t$ yields the correction $\mathcal{P}_T$.

**Definition 2** (Data-dependent gradient accumulation). *The* data-dependent gradient accumulation *is*

$$\mathcal{G}_T \;:=\; \sum_{t=1}^{T}\|\nabla f_t(x_t)\|^2. \tag{4}$$

*By Assumption 2, $\mathcal{G}_T \leq G_f^2 T$, with equality only if $\|\nabla f_t(x_t)\| = G_f$ at every round.*

These two quantities jointly determine the improvement of our refined bound over the prior bound. Corollary 4 (Section 5) shows that the improvement is $\Delta_T = \frac{\eta}{2}(G_f^2 T - \mathcal{G}_T) + \frac{1}{2\eta}\mathcal{P}_T \geq 0$, which is zero only in the degenerate case where all gradient norms attain their maximum and the Polyak step is never active.

## 4 Algorithm

We study the OGD with Polyak Feasibility Step (OGD-PFS) algorithm (Hutchinson & Alizadeh, 2025), restated in Algorithm 1. Our contribution is not a new algorithm but a refined analysis of this existing procedure; we include the full specification here for self-containedness.

Each round consists of two updates. The gradient step (Line 5) performs standard OGD: $y_t = x_t - \eta\nabla f_t(x_t)$, producing an intermediate iterate that may violate the constraint. The Polyak feasibility step (Lines 6–10) projects $y_t$ onto the half-space

$$H_t := \{x \in \mathbb{R}^d : g_t + s_t^\top(x - x_t) + \rho \leq 0\}, \tag{5}$$

which is a first-order outer approximation of the shrunk feasible set $\mathcal{X}_\rho$, and then clips the result to $R\mathcal{B}$. When $y_t$ already satisfies the linearized constraint ($y_t \in H_t$), the Polyak step reduces to the identity and $\delta_t = 0$. When $y_t \notin H_t$, the step moves $y_t$ by a squared distance of exactly $\delta_t = \|y_t - \Pi_{H_t}(y_t)\|^2 > 0$ back toward feasibility. The per-round cost is $O(d)$ plus one gradient evaluation and one constraint oracle call.

The shrinkage parameter $\rho \geq 0$ governs the feasibility–regret trade-off. Setting $\rho > 0$ enforces a tighter constraint $g(x) \leq -\rho$, providing a safety margin at the expense of an additional $G_f \rho T/\sigma$ term in the regret bound. Concrete parameter choices are given in Corollaries 1–3.

---

[1] We adopt the calligraphic symbol $\mathcal{P}_T$ (rather than the plain $P_T$) deliberately, to avoid clash with the standard dynamic-regret path-length notation $P_T = \sum_{t=1}^{T-1}\|x_t^\star - x_{t+1}^\star\|$ used in non-stationary OCO. The Polyak correction $\mathcal{P}_T$ here is a per-instance algorithmic quantity, not a path-length of comparators.

---

**Algorithm 1** OGD with Polyak Feasibility Step (OGD-PFS) (Hutchinson & Alizadeh, 2025)

---

**Require:** Initial point $x_1 \in R\mathcal{B}$, learning rate $\eta > 0$, shrinkage parameter $\rho \in [0, \epsilon]$
 1: **for** $t = 1, 2, \ldots, T$ **do**
 2:   Play action $x_t$; receive convex cost function $f_t$
 3:   Observe cost gradient $\nabla f_t(x_t)$
 4:   Query constraint oracle: $g_t \leftarrow g(x_t)$,   $s_t \leftarrow$ any element of $\partial g(x_t)$
 5:   **Gradient step:** $y_t \leftarrow x_t - \eta \nabla f_t(x_t)$
 6:   **if** $s_t = 0$ **then**
 7:     $x_{t+1} \leftarrow \Pi_{R\mathcal{B}}(y_t)$
 8:   **else**
 9:     **Polyak feasibility step:**
10:       $x_{t+1} \leftarrow \Pi_{R\mathcal{B}}\left( y_t - \dfrac{[g_t + s_t^\top (y_t - x_t) + \rho]_+}{\|s_t\|^2} s_t \right)$
11:   **end if**
12: **end for**

---

---

**Algorithm 2** Adaptive OGD with Polyak Feasibility Step (**AdaOGD-PFS**)

---

**Require:** Initial point $x_1 \in R\mathcal{B}$, shrinkage $\rho \in [0, \epsilon]$, scale $c > 0$, regularizer $\epsilon_0 > 0$
 1: Initialize $S_1 \leftarrow \epsilon_0$
 2: **for** $t = 1, 2, \ldots, T$ **do**
 3:   Play action $x_t$; receive convex cost function $f_t$
 4:   Observe cost gradient $\nabla f_t(x_t)$
 5:   Query constraint oracle: $g_t \leftarrow g(x_t)$,   $s_t \leftarrow$ any element of $\partial g(x_t)$
 6:   **Adaptive step size:** $\eta_t \leftarrow c / \sqrt{S_t}$
 7:   **Gradient step:** $y_t \leftarrow x_t - \eta_t \nabla f_t(x_t)$
 8:   **if** $s_t = 0$ **then**
 9:     $x_{t+1} \leftarrow \Pi_{R\mathcal{B}}(y_t)$
10:   **else**
11:     $x_{t+1} \leftarrow \Pi_{R\mathcal{B}}\left( y_t - \dfrac{[g_t + s_t^\top (y_t - x_t) + \rho]_+}{\|s_t\|^2} s_t \right)$
12:   **end if**
13:   $S_{t+1} \leftarrow S_t + \|\nabla f_t(x_t)\|^2$
14: **end for**

---

## 4.1  Adaptive OGD with Polyak Feasibility Steps

Our refined analysis (Theorem 1) reveals that the actual gradient accumulation $\mathcal{G}_T$ and the Polyak correction $\mathcal{P}_T$ are data-dependent quantities that can be much more favorable than their worst-case surrogates. A natural question is whether an algorithm can *exploit* this data-dependence online, rather than merely observing it post-hoc.

We answer this affirmatively by introducing **AdaOGD-PFS** (Algorithm 2), which replaces the fixed step size $\eta$ with an adaptive step size $\eta_t$ that shrinks as gradient information accumulates, in the spirit of AdaGrad (Duchi et al., 2011), while retaining the Polyak feasibility step unchanged.

The key difference from Algorithm 1 is Line 6: the step size $\eta_t = c / \sqrt{S_t}$ decreases as the cumulative squared gradient norm $S_t = \epsilon_0 + \sum_{i=1}^{t-1} \|\nabla f_i(x_i)\|^2$ grows. When the cost gradients are small on average ($\mathcal{G}_T \ll G_f^2 T$), the step sizes remain larger for longer, allowing the algorithm to make more aggressive updates. When gradients are large, the step sizes shrink rapidly, maintaining stability. The Polyak feasibility step (Lines 8–11) is identical to Algorithm 1.

The hyperparameter $c > 0$ controls the step size scale; the choice $c = R\sqrt{2}$ minimizes the leading constant in the regret bound (Theorem 2). The regularizer $\epsilon_0 > 0$ ensures that $\eta_1 = c/\sqrt{\epsilon_0}$ is finite and, when chosen appropriately, guarantees per-round feasibility (Corollary 5).

## 5 Main Results

This section presents our theoretical contributions. Section 5.1 gives the refined analysis of the fixed-step-size algorithm (Theorem 1), which tightens the prior bound without modifying the algorithm. Section 5.5 gives the analysis of AdaOGD-PFS (Theorem 2), which achieves a fully data-dependent regret bound algorithmically. All proofs are deferred to Appendix A; a proof overview is given in Section 6.

### 5.1 Refined Regret Bound and Feasibility Guarantee

**Theorem 1** (Refined regret bound and feasibility guarantee). *Let Assumptions 1–4 hold. Run Algorithm 1 with $x_1 \in R\mathcal{B}$, $\eta > 0$, and $\rho \in [0, \epsilon]$.*

*(I) Regret bound:*

$$\mathrm{Reg}_T \ \leq \ \frac{2R^2}{\eta} + \frac{\eta}{2}\,\mathcal{G}_T - \frac{1}{2\eta}\,\mathcal{P}_T + \frac{G_f \rho}{\sigma}\,T\,. \tag{6}$$

*(II) Feasibility bound: For all $t \geq 1$,*

$$g(x_t) \ \leq \ G_g\,\gamma^{(t-1)/2}\,\mathrm{dist}(x_1, \mathcal{X}_\rho) + \frac{\eta G_g G_f}{\xi} - \rho\,. \tag{7}$$

The proof is given in Appendix A.6. Compared to the prior bound $\frac{2R^2}{\eta} + \frac{\eta}{2}G_f^2 T + \frac{G_f \rho}{\sigma}T$, the regret bound equation 6 is tighter in two ways: $\mathcal{G}_T \leq G_f^2 T$ (data-dependent gradient term) and $-\frac{1}{2\eta}\mathcal{P}_T \leq 0$ (Polyak correction, absent from the prior bound). The feasibility bound equation 7 is identical to that of prior work.

### 5.2 Corollaries

All corollary proofs are given in Appendix A.7.

**Corollary 1** (Per-round feasibility with known strictly feasible point). *Suppose $g(x_1) \leq -\alpha$ for some $\alpha \in (0, \epsilon]$. Set $\rho = \alpha/\sqrt{T}$ and $\eta = \xi\alpha/(G_f G_g \sqrt{T})$. Then $g(x_t) \leq 0$ for all $t \in [T]$, and*

$$\mathrm{Reg}_T \leq \left( \frac{2G_g R^2}{\xi\alpha} + \frac{\xi\alpha}{2G_g} \cdot \frac{\mathcal{G}_T}{G_f^2 T} + \frac{\alpha}{\sigma} \right) G_f \sqrt{T} - \frac{G_f G_g \sqrt{T}}{2\xi\alpha}\,\mathcal{P}_T\,. \tag{8}$$

This corollary treats the practical case where the learner knows a strictly feasible starting point with margin $\alpha$. The parameters $(\rho, \eta)$ are picked so that the shrinkage is just large enough to absorb the residual feasibility error in equation 7, yielding $g(x_t) \leq 0$ at every round. The three positive terms inside the bracket are non-monotone in $\alpha$: a larger margin shrinks the first term $2G_g R^2/(\xi\alpha)$ but inflates the second ($\propto \alpha\mathcal{G}_T/(G_f^2 T)$) and third ($\alpha/\sigma$) terms, while a smaller margin does the opposite; the bracket as a function of $\alpha$ is minimised at an interior $\alpha^\star$ that balances these effects. The Polyak term enters with a strictly negative sign whose coefficient $G_f G_g \sqrt{T}/(2\xi\alpha)$ grows when $\alpha$ is small, so tight margins amplify both the cost from the first term and the benefit from the Polyak correction.

**Corollary 2** (Delayed feasibility with unknown interior point). *Set $\rho = \epsilon/\sqrt{T}$ and $\eta = \xi\epsilon/(2G_f G_g \sqrt{T})$. Then $g(x_t) \leq 0$ for all $t \geq 1 + \frac{2G_g^2}{\sigma^2}\log\left(\frac{4G_g R\sqrt{T}}{\epsilon}\right)$, and*

$$\mathrm{Reg}_T \leq \left( \frac{4G_f G_g R^2}{\xi\epsilon} + \frac{\xi\epsilon G_f}{4G_g} + \frac{G_f \epsilon}{\sigma} \right) \sqrt{T} - \frac{G_f G_g \sqrt{T}}{\xi\epsilon}\,\mathcal{P}_T\,. \tag{9}$$

*Moreover, $\sum_{t=1}^T g(x_t) \leq 0$ whenever $\sqrt{T} \geq 4RG_g/(\epsilon\xi)$.*

The setting here covers the case where no strictly feasible point is known a priori, only the Slater margin $\epsilon$. Per-round feasibility is then guaranteed only after a logarithmic burn-in of order $(G_g/\sigma)^2 \log T$, during which the iterate is pulled toward $\mathcal{X}_\rho$ by repeated Polyak steps; this transient is the price paid for not

knowing a feasible warm start. The cumulative-feasibility statement is a useful by-product: as soon as $\sqrt{T}$ exceeds a problem-dependent constant, the running sum of constraint values is non-positive, so any sublinear cumulative-violation metric is automatically tight.

**Corollary 3** (No shrinkage). *Set $\rho = 0$ and $\eta = 2R/(G_f\sqrt{T})$. Then*

$$\text{Reg}_T \le RG_f\sqrt{T} + \frac{R\,\mathcal{G}_T}{G_f\sqrt{T}} - \frac{G_f\sqrt{T}}{4R}\,\mathcal{P}_T\,, \tag{10}$$

*and $g(x_t) \le 2RG_g e^{-\sigma^2(t-1)/(2G_g^2)} + 2RG_g/(\xi\sqrt{T})$ for all $t$.*

This is the cleanest instantiation for the experiments: with $\rho = 0$ and the standard step $\eta = 2R/(G_f\sqrt{T})$ the first two terms reduce to $RG_f\sqrt{T} + R\mathcal{G}_T/(G_f\sqrt{T})$, which is the data-dependent counterpart of the prior bound $2RG_f\sqrt{T}$, and they coincide only when $\mathcal{G}_T = G_f^2 T$. The third term $-G_f\sqrt{T}\,\mathcal{P}_T/(4R)$ contributes a strictly negative correction whenever the Polyak step is active. The feasibility bound exhibits the typical two-phase behaviour: an exponentially decaying transient driven by the contraction factor $\sigma^2/G_g^2$, followed by a residual $O(1/\sqrt{T})$ steady state.

### 5.3  Improvement Decomposition

**Corollary 4** (Improvement decomposition). *The improvement of equation 6 over the prior bound is*

$$\Delta_T := \underbrace{\frac{\eta}{2}(G_f^2 T - \mathcal{G}_T)}_{gradient\ refinement\ \ge\ 0} + \underbrace{\frac{1}{2\eta}\mathcal{P}_T}_{Polyak\ correction\ \ge\ 0} \ge 0\,. \tag{11}$$

*$\Delta_T = 0$ iff $\|\nabla f_t(x_t)\| = G_f$ for all $t$ and $\mathcal{P}_T = 0$.*

The two components of $\Delta_T$ are functionally orthogonal: the gradient-refinement term $\frac{\eta}{2}(G_f^2 T - \mathcal{G}_T)$ is an *environment* property and is large precisely when the cost gradients are heterogeneous so that $\mathcal{G}_T \ll G_f^2 T$; the Polyak-correction term $\frac{1}{2\eta}\mathcal{P}_T$ is an *algorithm-plus-environment* property and is large precisely when the gradient step frequently pushes the iterate outside the feasible half-space. The two components also depend oppositely on $\eta$: a larger step makes the gradient-refinement coefficient ($\eta/2$) larger but the Polyak-correction coefficient ($1/(2\eta)$) smaller, while $\mathcal{P}_T$ itself tends to grow with $\eta$ since larger steps create larger linearised violations. The corner case $\Delta_T = 0$ requires both sources to vanish simultaneously, i.e., the cost is uniformly worst-case (so the first term is zero) *and* the gradient never pushes the iterate outside the half-space (so $\mathcal{P}_T = 0$); this is the only configuration in which our bound exactly matches the prior bound.

### 5.4  Properties of the Polyak Correction

The proofs of Propositions 1 and 2 are given in Appendix A.7.

**Proposition 1** (Lower bound on Polyak correction). *Let $\mathcal{A} := \{t \in [T] : g_t + s_t^\top(y_t - x_t) + \rho > 0\}$ be the set of active rounds. Then*

$$\mathcal{P}_T = \sum_{t\in\mathcal{A}} \frac{(g_t + s_t^\top(y_t - x_t) + \rho)^2}{\|s_t\|^2} \ge \frac{1}{G_g^2}\sum_{t\in\mathcal{A}}(g_t + s_t^\top(y_t - x_t) + \rho)^2. \tag{12}$$

This lower bound shows that the Polyak correction is summed only over the active set $\mathcal{A}$ and that each summand is at least the squared linearised violation divided by the worst-case subgradient norm $G_g^2$. The active fraction $|\mathcal{A}|/T$ is therefore the natural complexity measure of how often the Polyak step does corrective work; we report it in every experimental row of Tables 3 and 5 and discuss its interaction with $\mathcal{P}_T$ in Section 7.4.

**Proposition 2** (Extreme cases). *(a) If $g_t + s_t^\top(y_t - x_t) + \rho \le 0$ for all $t$, then $\mathcal{P}_T = 0$ and $\text{Reg}_T \le \frac{2R^2}{\eta} + \frac{\eta}{2}\mathcal{G}_T + \frac{G_f\rho}{\sigma}T$. (b) If $g_t + s_t^\top(y_t - x_t) + \rho = c > 0$ for all $t$, then $\mathcal{P}_T \ge c^2 T/G_g^2$ and the Polyak correction yields an $O(T)$-magnitude reduction $-c^2 T/(2\eta G_g^2)$.*

The two extreme cases above delineate the qualitative regimes the Polyak term can land in. Case (a) is the strictly-interior regime, where the gradient step never crosses the linearised constraint and the bound reduces to the data-dependent counterpart of the standard projected-OGD bound; this is the only setting where the Polyak correction can vanish despite the algorithm being nominally Polyak-augmented. Case (b) is the maximally-active regime, where the Polyak step performs a positive amount of work at every round; the reduction is then linear in $T$ and can be the dominant component of $\Delta_T$. The adversarial experiment in Section 7.5 is exactly an instance close to case (b) and explains why the Polyak correction there reaches 49.9% of the prior bound.

Beyond the post-hoc numerical tightening reported in the experiments, the Polyak correction $\mathcal{P}_T$ admits three concrete uses that clarify its practical and theoretical role. Since both $\mathcal{G}_T$ and $\mathcal{P}_T$ are computed online at zero extra oracle cost ($\delta_t$ is a by-product of the Polyak step), the quantity $\widehat{\Delta}_t = \frac{\eta}{2}(G_f^2 t - \mathcal{G}_t) + \frac{1}{2\eta}\mathcal{P}_t$ is an anytime per-trajectory certificate: at any round $t$ the deployment knows that its regret is upper-bounded by the prior bound minus $\widehat{\Delta}_t$, which can be reported to the user as a real-time tightness gap, making the bound auditable in safety-critical deployments. The same quantity also yields a qualitative change of guarantee on highly-active instances: on problem classes for which one can verify a priori that the linearised constraint will be active in a constant fraction of rounds with violation bounded below by a constant $c > 0$ (as in case (b) of Proposition 2), the Polyak correction satisfies $\mathcal{P}_T \geq c^2 T / G_g^2 = \Omega(T)$, and substituting into Corollary 3 with $\eta = 2R/(G_f\sqrt{T})$ makes the negative term $-\frac{G_f\sqrt{T}}{4R}\mathcal{P}_T = -\Omega(T^{3/2})$ exceed the $O(\sqrt{T})$ positive terms in absolute value, so the prior $\Theta(\sqrt{T})$-style guarantee is no longer tight on this subclass; the adversarial experiment in Section 7.5 ($\text{Reg}_T = 74$ at $T = 10{,}000$, empirically constant-scale and far below the $\Theta(\sqrt{T})$ envelope) is exactly this regime, and whether $\text{Reg}_T = O(1)$ holds rigorously on this subclass is left as a corollary-style consequence. Finally, the same online-computable ratio $\mathcal{P}_t/(G_f^2 t)$ doubles as a tuning diagnostic: it tells the user whether the current step size is in the over-cautious regime ($\mathcal{P}_t \approx 0$; the gradient step never crosses the linearised constraint, so $\eta$ is too small) or the over-aggressive regime ($\mathcal{P}_t/(G_f^2 t)$ saturating; the algorithm frequently overshoots), so that a simple doubling-trick wrapper on $\eta$ monitoring this ratio can certify a near-optimal step-size choice on the executed trajectory without re-running the algorithm. We do not formalise such a wrapper in this paper but flag it as a concrete tuning use of $\mathcal{P}_T$.

## 5.5 Data-Dependent Bound for AdaOGD-PFS

**Theorem 2** (Data-dependent regret bound for AdaOGD-PFS)**.** *Let Assumptions 1–4 hold. Run Algorithm 2 with $x_1 \in R\mathcal{B}$, $\rho \in [0, \epsilon]$, $c > 0$, and $\epsilon_0 > 0$.*

*(I) Regret bound:*

$$\text{Reg}_T \;\leq\; \left(\frac{2R^2}{c} + c\right)\sqrt{\mathcal{G}_T + \epsilon_0} \;+\; \frac{c\,G_f^2}{2\sqrt{\epsilon_0}} \;-\; \sum_{t=1}^{T}\frac{\delta_t}{2\eta_t} \;+\; \frac{G_f\rho}{\sigma}\,T\,. \tag{13}$$

*With the optimal scale $c = R\sqrt{2}$, the first term becomes $2R\sqrt{2(\mathcal{G}_T + \epsilon_0)}$. The additional term $cG_f^2/(2\sqrt{\epsilon_0})$ comes from the past-gradient denominator $\eta_t = c/\sqrt{S_t}$ used in Algorithm 2, where $S_t$ excludes the current squared gradient $\|\nabla f_t(x_t)\|^2$; an alternative algorithm that uses the current-gradient denominator $\eta_t = c/\sqrt{S_t + \|\nabla f_t(x_t)\|^2}$ removes this term while leaving the leading $O(\sqrt{\mathcal{G}_T})$ rate unchanged (cf. Remark 2).*

*(II) Feasibility bound: For all $t \geq 1$,*

$$g(x_t) \;\leq\; G_g\,\gamma^{(t-1)/2}\,\text{dist}(x_1, \mathcal{X}_\rho) + \frac{c\,G_g G_f}{\xi\sqrt{\epsilon_0}} - \rho\,. \tag{14}$$

The proof is given in Appendix A.8. The regret bound equation 13 replaces the $O(G_f\sqrt{T})$ dependence of both the prior bound and Theorem 1 with $O(\sqrt{\mathcal{G}_T})$, which is never worse and can be substantially better when the cost gradients along the algorithm's trajectory are heterogeneous. The Polyak correction term

$-\sum_t \delta_t/(2\eta_t)$ now has a time-varying coefficient $1/(2\eta_t) = \sqrt{S_t}/(2c)$ that *increases* over time, weighting later corrections more heavily than in the fixed-step-size analysis.

**Remark 2** (Past-gradient vs. current-gradient denominator)**.** *Algorithm 2 uses the standard past-gradient AdaGrad denominator $\eta_t = c/\sqrt{S_t}$ with $S_t = \epsilon_0 + \sum_{i<t}\|\nabla f_i(x_i)\|^2$, which keeps $\eta_t$ measurable with respect to information available* before *the round-t play. Because the denominator excludes the current $\|\nabla f_t(x_t)\|^2$, the AdaGrad telescoping in step (B) of the proof of Theorem 2 (Appendix A.8) incurs an additional $cG_f^2/(2\sqrt{\epsilon_0})$ overhead beyond the leading $c\sqrt{\mathcal{G}_T + \epsilon_0}$ term. This overhead is a constant (independent of T) and is dominated by the $O(\sqrt{T})$ leading term, so the asymptotic $O(\sqrt{\mathcal{G}_T})$ rate is unaffected. If one prefers a bound without this overhead, replacing the algorithmic denominator with the current-gradient version $\eta_t = c/\sqrt{S_t + \|\nabla f_t(x_t)\|^2}$ removes the term entirely; the resulting algorithm is no longer "measurable-at-time-t − 1" but is still well-defined since $\nabla f_t(x_t)$ is observed before $\eta_t$ is used to update.*

**Corollary 5** (AdaOGD-PFS with per-round feasibility)**.** *Suppose $g(x_1) \le -\alpha$ for some $\alpha \in (0, \epsilon]$. Set $c = R\sqrt{2}$, $\rho = \alpha/\sqrt{T}$, and $\epsilon_0 = 2G_g^2 G_f^2 R^2/(\xi^2 \rho^2) = 2G_g^2 G_f^2 R^2 T/(\xi^2 \alpha^2)$. Then $g(x_t) \le 0$ for all $t \in [T]$, and*

$$\text{Reg}_T \le 2R\sqrt{2\mathcal{G}_T + 2\epsilon_0} - \sum_{t=1}^{T} \frac{\delta_t}{2\eta_t} + \frac{G_f \alpha}{\sigma}\sqrt{T}. \tag{15}$$

*(The past-gradient overhead $cG_f^2/(2\sqrt{\epsilon_0})$ from Theorem 2 evaluates, under this parameter choice, to $\xi\alpha G_f/(2G_g\sqrt{T}) = O(1/\sqrt{T})$, which is absorbed into the leading $\sqrt{T}$ term and is therefore omitted from equation 15; see Remark 2.)*

**Remark 3** (Scope of the "no knowledge of $G_f$" claim)**.** *We make this trade-off explicit. The regret bound of Theorem 2 for AdaOGD-PFS truly does* not *require knowledge of $G_f$: the algorithm is run with $\eta_t = c/\sqrt{S_t}$ using only the cost gradients observed online, and the bound $(2R^2/c + c)\sqrt{\mathcal{G}_T + \epsilon_0}$ scales in $\mathcal{G}_T$ with constants that do not contain $G_f$. The per-round feasibility guarantee in Corollary 5, however, is established by tuning $\epsilon_0$ so that the worst-case step size $\eta_1 = c/\sqrt{\epsilon_0}$ is small enough; the specific choice $\epsilon_0 = 2G_g^2 G_f^2 R^2/(\xi^2 \alpha^2) \cdot T$ does depend on $G_f$, $G_g$, $\xi$ and $\alpha$. Thus the slogan "$O(\sqrt{\mathcal{G}_T})$ regret without knowing $G_f$" applies to the* regret rate, *while per-round* feasibility under our analysis still uses $G_f$ to set the regularizer. When $G_f$ is unknown, one obtains per-round feasibility only up to the same uniform-$G_f$ slack as the fixed-step algorithm. We make this explicit so that the reader is not misled into expecting feasibility guarantees that are themselves $G_f$-free, and we explicitly disclaim that this paper provides a $G_f$-free alternative parameter choice that simultaneously achieves $O(\sqrt{\mathcal{G}_T})$ regret and per-round feasibility; constructing such a doubly $G_f$-free schedule (perhaps via a doubling trick on an online estimate of $G_f$) is left for future work.*

The progression from the prior bound to Theorem 1 and subsequently to Theorem 2 reflects two distinct advancements: developing a tighter analysis for the existing algorithm, followed by the introduction of a new algorithm with a data-dependent rate. Each step constitutes a strict improvement, and a comprehensive comparison is provided in Table 1.

## 6  Proof Overview

All proofs are deferred to Appendix A. Here we summarize the proof architecture for Theorem 1. The argument proceeds via five supporting lemmas. Lemma 1 provides the enhanced projection inequality that retains $\delta_t$, which is the slack in the strong projection inequality (Bauschke & Combettes, 2017, Cor. 4.10) that the standard worst-case OCO argument does not need to track: for all $x \in \mathcal{X}_\rho$, $\|x_{t+1} - x\|^2 \le \|y_t - x\|^2 - \delta_t$. Lemma 2 is a constraint error bound relating $\text{dist}(x, \mathcal{X}_\rho)$ to $[g(x) + \rho]_+/\sigma$. Lemma 3 establishes geometric contraction $\text{dist}^2(x^+, \mathcal{X}_\rho) \le \gamma \cdot \text{dist}^2(x, \mathcal{X}_\rho)$ of the Polyak step. Lemmas 4 and 5 develop the feasibility recursion and its expansion. The regret bound then follows by a telescoping argument that, unlike the standard analysis, retains both $\delta_t$ and $\|\nabla f_t(x_t)\|^2$ without relaxation. The proof of Theorem 2 adapts this telescoping to time-varying step sizes via Abel summation.

Table 3: Experimental results ($T = 10{,}000$, $d = 10$, $\rho = 0$, means over 5 seeds). "Prior" is the bound of Hutchinson & Alizadeh (2025); "Thm. 1" is our refined bound; "Thm. 2" is the AdaOGD-PFS bound. All bounds are numerically verified to upper-bound the actual regret in every run. The $\mathrm{Reg}_T$ column reports the actual regret of OGD-PFS (Algorithm 1). The corresponding actual regret of AdaOGD-PFS (Algorithm 2) is reported separately in Appendix B: 373 (Ball+Linear), 1,173 (Ball+Linear HT), and 148 (Halfspace); both algorithms remain well below their respective bounds.

| **Problem** | $\mathrm{Reg}_T$ | **Prior** | **Thm. 1** | **Thm. 2** | $\mathcal{G}_T/(G_f^2 T)$ | $|\mathcal{A}|/T$ |
|---|---|---|---|---|---|---|
| Ball+Linear | 235 | 3,541 | 2,089 | 2,150 | 0.281 | 0.969 |
| Ball+Linear (HT) | 643 | 7,486 | 4,650 | 5,164 | 0.270 | 0.637 |
| Halfspace | 173 | 3,897 | 2,239 | 2,295 | 0.314 | 0.994 |

## 7 Experiments

We validate the theoretical results on synthetic constrained OCO instances and verify that the refined bounds (Theorems 1 and 2) are valid upper bounds on the actual regret, that the improvement over the prior bound is substantial and robust, and that AdaOGD-PFS achieves competitive actual regret compared to fixed-step-size OGD-PFS.

### 7.1 Setup

**Problem instances.** We consider three constrained OCO problems, all with dimension $d = 10$, bounding radius $R = 5$, and time horizon $T = 10{,}000$. The first configuration applies a ball constraint $g(x) = \|x\|^2/r^2 - 1$ with $r = 2$ to linear costs $f_t(x) = a_t^\top x$, where the cost vectors are generated as $a_t = \bar{a} + \epsilon_t$ with $\bar{a} = e_1$ and $\epsilon_t \sim \mathcal{N}(0, 0.25\, I_d)$. A heavy-tailed variant of this setting uses the identical constraint but samples perturbations from $\epsilon_t \sim \mathcal{N}(0, 1.44\, I_d)$, which introduces occasional large gradient norms that inflate the worst-case bound $G_f$. The final configuration features a halfspace constraint $g(x) = e_1^\top x - 2$ paired with linear costs, where the mean component of the cost vector continuously pushes the sequence toward the feasibility boundary.

**Methods.** For each problem, we compare two methods. The first is OGD-PFS (Algorithm 1), configured with $\eta = 2R/(G_f\sqrt{T})$ and $\rho = 0$, which we evaluate under both the prior bound and our refined bound (Theorem 1). The second method is AdaOGD-PFS (Algorithm 2), configured with $c = R\sqrt{2}$, $\epsilon_0 = G_f^2$, and $\rho = 0$, which is evaluated under Theorem 2. All results are averaged over five random seeds. To avoid favouring our bound through an inflated worst-case constant, in every run we set $G_f := \max_{t \in [T]} \|\nabla f_t(x_t)\|$, i.e., the smallest scalar that is a valid Lipschitz envelope along the executed trajectory. Consequently the prior bound is evaluated at its tightest possible value of $G_f$, and any reported improvement is a strict tightening on top of the best achievable worst-case constant.

### 7.2 Bound Validity and Improvement

Table 3 reports the actual regret, the three bounds, and the key diagnostic quantities for each problem instance.

Several observations are noteworthy. First, the bound validity is confirmed: the refined bounds (Theorems 1 and 2) are valid upper bounds on $\mathrm{Reg}_T$ in all runs, a necessary sanity check for the theoretical analysis. Second, both refined bounds improve substantially over the prior: Theorem 1 achieves a 38–43% reduction, while Theorem 2 achieves a 31–41% reduction. Third, the data-dependent gradient ratio $\mathcal{G}_T/(G_f^2 T)$ ranges from 0.27 to 0.31, confirming that the worst-case gradient constant $G_f$ significantly overestimates the actual gradient magnitudes. The active fraction $|\mathcal{A}|/T$ ranges from 0.64 to 0.99, indicating that the Polyak step is frequently invoked. Fourth, and to be transparent: in every reported instance the AdaOGD-PFS bound (column Thm. 2 in Table 3: 2150, 5164, 2295) is slightly larger than the fixed-step Theorem 1 bound (2089, 4650, 2239), because the fixed step $\eta = 2R/(G_f\sqrt{T})$ is tuned with oracle knowledge of $G_f$. The intended advantage of AdaOGD-PFS is therefore not to dominate Theorem 1 numerically when $G_f$ is known and

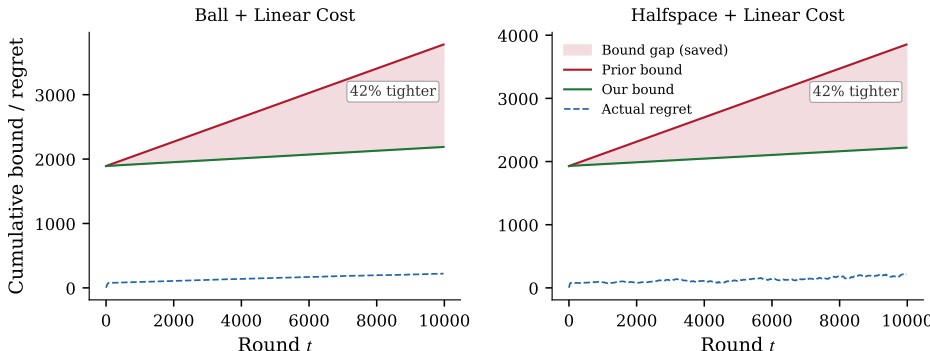

Figure 1: Cumulative regret and bounds over $T = 10,000$ rounds for ball-constrained (left) and halfspace-constrained (right) problems. The shaded region between the prior bound (red) and our refined bound (green) represents the improvement from Theorem 1. Both bounds are valid upper bounds on the actual regret (blue dashed). The "actual regret" curve in both panels is that of OGD-PFS (Algorithm 1). The actual regret of AdaOGD-PFS (Algorithm 2) is omitted from these panels for visual clarity and is reported numerically in Table 3 and discussed in Appendix B; Figure 2 provides a three-way visual comparison.

Table 4: Decomposition of the bound improvement (Corollary 4). Values are mean $\pm$ std over 5 seeds.

| Problem | Gradient (%) | Polyak (%) | Total (%) |
|---|---|---|---|
| Ball+Linear | $35.9 \pm 1.2$ | $5.0 \pm 0.4$ | $41.0 \pm 0.8$ |
| Ball+Linear (HT) | $36.5 \pm 1.2$ | $1.4 \pm 0.1$ | $37.8 \pm 1.1$ |
| Halfspace | $34.3 \pm 1.0$ | $8.2 \pm 0.5$ | $42.5 \pm 0.5$ |

oracle-tuned, but to obtain an $O(\sqrt{\mathcal{G}_T})$ rate (rather than $O(G_f\sqrt{T})$) with constants that are independent of $G_f$ in the leading term; this advantage is most visible when $G_f$ is unknown or substantially overestimated. See Remark 3 for the precise scope of this claim.

### 7.3 Improvement Decomposition

Table 4 decomposes the Theorem 1 improvement into its two components.

The gradient refinement ($\mathcal{G}_T$ replacing $G_f^2 T$) is the dominant source of improvement, contributing 34–37% across all problems. The Polyak correction ($\mathcal{P}_T$) provides an additional 1–8%, with the largest contribution on the halfspace problem where the constraint is active in 99.4% of rounds. Both components are strictly positive in all instances, consistent with the theoretical guarantee $\Delta_T \geq 0$ (Corollary 4). Additional experiments are presented in Appendix B. These include a comparison of AdaOGD-PFS versus fixed-step-size OGD-PFS, bound component decomposition, sensitivity analysis over $(\eta, \rho)$, and distributional analysis of gradient norms and Polyak corrections.

### 7.4 Active Fraction and Its Connection to the Polyak Correction

By Proposition 1, the Polyak correction can be written as a sum over the active set $\mathcal{A} = \{t : g_t + s_t^\top(y_t - x_t) + \rho > 0\}$, so the active fraction $|\mathcal{A}|/T$ is the natural "how often does the Polyak step do non-trivial work" diagnostic. Empirically the two quantities track each other but with an instance-dependent slope. On the halfspace problem the active fraction is highest (0.994) and so is the Polyak share of the improvement (8.2%); on the heavy-tailed ball problem the active fraction is the lowest (0.637) and the Polyak share drops to 1.4%; the standard ball problem sits in between (0.969 active fraction, 5.0% Polyak share).

Beyond this rough monotone trend, the active fraction alone does *not* determine the Polyak share, because the per-round displacement $\delta_t$ depends on the squared linearised violation $(g_t + s_t^\top(y_t - x_t) + \rho)^2$, not just on its sign. On the heavy-tailed ball instance the active fraction is moderate but the typical $\delta_t$ is small, because

Table 5: Adversarial worst-case experiment. Setup is identical to the Ball+Linear row of Table 3 except the cost gradient is held at the worst case $\nabla f_t(x_t) = G_f e_1$ for every $t$, so $\mathcal{G}_T = G_f^2 T$ exactly and the gradient-refinement contribution is zero by construction. Five seeds; the refined bound remains a valid upper bound on the actual regret in every run.

| Setting | $\text{Reg}_T$ | Prior | Thm. 1 | Grad. (%) | Polyak (%) | Total (%) |
|---|---|---|---|---|---|---|
| Gaussian (paper setup, sanity) | 235 | 3,541 | 2,089 | 35.9 | 5.0 | 41.0 |
| Adversarial $\nabla f_t = G_f e_1 \ \forall t$ | 74 | 3,541 | 1,774 | 0.0 | 49.9 | 49.9 |

the heavy-tailed noise randomises the direction of $y_t - x_t = -\eta \nabla f_t(x_t)$ across rounds, so the alignment $s_t^\top(y_t - x_t)$ on the constraint normal is often small even when $g_t$ alone would trigger an active round; this explains the disproportionately small Polyak share there.

A related observation is that the active fraction is essentially uncorrelated with how the *algorithm itself* performs across instances. The two ball problems have very different active fractions (0.97 vs. 0.64) but comparable OGD-PFS regret normalised by $G_f\sqrt{T}$, while the halfspace instance has the highest active fraction but the lowest absolute regret. The active fraction therefore predicts how much of the Polyak *correction* is excited, not how well either algorithm does in terms of actual regret. The adversarial experiment in Section 7.5 makes the same point quantitatively: pushing the active fraction to 0.998 multiplies the Polyak share roughly tenfold ($5.0\% \to 49.9\%$) while leaving the prior bound unchanged.

### 7.5 Adversarial Worst-Case Gradient Sequence

A natural concern is whether the 38–43% headline improvement merely reflects Gaussian concentration of $\|\nabla f_t\|$ below $G_f$, in which case any AdaGrad-style analysis would extract the same gain. To isolate the contribution of the Polyak correction $\mathcal{P}_T$ from that of the gradient refinement, we add a strictly adversarial experiment using the same Ball+Linear configuration ($T = 10{,}000$, $d = 10$, $R = 5$, $r = 2$, $\rho = 0$, $\eta = 2R/(G_f\sqrt{T})$, five seeds, identical to Table 3), except that the cost gradient is held constant at the worst case $\nabla f_t(x_t) = G_f e_1$ for every round. Under this sequence $\mathcal{G}_T = G_f^2 T$ exactly, so the gradient-refinement term contributes *zero* improvement by construction; any remaining gap is attributable solely to the Polyak correction. Table 5 reports the result.

Two facts are worth highlighting. First, the gradient-refinement contribution drops exactly to 0%, as predicted (in the Gaussian baseline it was 35.9%). This confirms that the gradient gain on the Gaussian benchmark is essentially a concentration effect that any data-dependent gradient analysis would capture. Second, and more importantly, the Polyak correction does *not* vanish under the adversarial sequence; it remains strictly positive and in fact *larger* than in the Gaussian case (49.9% vs. 5.0%). The reason is that the constant adversarial push keeps the iterate at the boundary in essentially every round (active fraction $\approx 0.998$), and the per-round Polyak displacement is also amplified by the smaller $\eta$ entering the coefficient $1/(2\eta)$. The refined bound is therefore strictly tighter than the prior bound by 49.9% *even when the gradient-refinement term is zero*, demonstrating that the Polyak correction $\mathcal{P}_T$ contributes a tightening that is structurally independent of, and not subsumed by, any AdaGrad-style gradient adaptation.

## 8 Conclusion

We have presented a refined regret analysis of OGD with Polyak feasibility steps in constrained online convex optimization. Our analysis identifies two quantities that the standard worst-case argument does not track, namely the data-dependent gradient accumulation and the Polyak correction, and retains both as data-dependent quantities. The resulting bound is uniformly no worse than the prior state-of-the-art and achieves 38–43% tighter bounds in experiments. Motivated by these analytical insights, we further propose AdaOGD-PFS, a new algorithm with adaptive step sizes that achieves an $O(\sqrt{\mathcal{G}_T})$ regret bound while maintaining per-round feasibility. Several directions for future work emerge from the current limitations. Because $\mathcal{P}_T$ is a post-hoc quantity evaluated after execution, the resulting bounds are inherently instance-dependent rather than worst-case guarantees. Furthermore, the feasibility bound remains unchanged from

prior literature, and tightening it via the actual gradient norms $\|\nabla f_t(x_t)\|$ instead of the worst-case $G_f$ presents a valuable theoretical opportunity. Another unresolved challenge is integrating the Polyak feasibility step with optimistic online gradient descent , as the standard optimistic telescoping introduces cross terms that are difficult to control.

Beyond the limitations of the present setting, the two devices used in our analysis are not specific to OGD with a single Polyak feasibility step, which suggests several natural extensions. The data-dependent gradient quantity $\mathcal{G}_T = \sum_t \|\nabla f_t(x_t)\|^2$ arises whenever one telescopes a Bregman-style regret identity without invoking the uniform bound $\|\nabla f_t(x_t)\| \leq G_f$ at the last step; it is therefore expected to be retainable in mirror descent, online Newton step, and Frank–Wolfe variants with linearised constraints, as well as in stochastic, bandit, and delayed-feedback settings where each round still produces a single gradient sample. The Polyak correction $\mathcal{P}_T = \sum_t \delta_t$, on the other hand, is the cumulative slack in the strong projection inequality, so an analogous correction can be defined whenever the algorithm contains a projection or proximal step onto a convex set: in particular, projected OGD onto an arbitrary convex set, alternating projection and Dykstra-style schemes, multi-constraint Polyak steps (one half-space per constraint per round), and constrained stochastic and bandit OCO. In each of these settings the slack term can be summed and subtracted from the regret bound, yielding a tightening that is structurally independent of any AdaGrad-style gradient adaptation. Quantifying these instantiations is left to future work.

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

## A  Proofs

### A.1  Lemma 1: Enhanced Polyak Projection Property

**Lemma 1** (Enhanced Polyak projection property). *Let Assumptions 1–4 hold and $\rho \in [0, \epsilon]$. Then Step 9 of Algorithm 1 satisfies: for all $x \in \mathcal{X}_\rho$,*

$$\|x_{t+1} - x\|^2 \leq \|y_t - x\|^2 - \delta_t. \tag{16}$$

*Proof.* **Case $s_t = 0$:** Since $0 \in \partial g(x_t)$, $x_t$ is a global minimizer of $g$. By Assumption 4, $\mathcal{X}_\rho \neq \emptyset$ (there exists $w$ with $g(w) = -\epsilon \leq -\rho$), so $g(x_t) \leq g(w) \leq -\rho$, i.e., $x_t \in \mathcal{X}_\rho$. The algorithm gives $x_{t+1} = \Pi_{R\mathcal{B}}(y_t)$ and $\delta_t = 0$. Since $x \in \mathcal{X}_\rho \subseteq R\mathcal{B}$, by non-expansiveness of $\Pi_{R\mathcal{B}}$: $\|x_{t+1} - x\|^2 \leq \|y_t - x\|^2 = \|y_t - x\|^2 - \delta_t$.

**Case $s_t \neq 0$:** We first verify that the algorithm step is equivalent to $x_{t+1} = \Pi_{R\mathcal{B}}(\Pi_{H_t}(y_t))$. If $g_t + s_t^\top(y_t - x_t) + \rho \leq 0$, then $y_t \in H_t$, so $\Pi_{H_t}(y_t) = y_t$ and the equivalence is immediate. If $g_t + s_t^\top(y_t - x_t) + \rho > 0$, then $\Pi_{H_t}(y_t) = y_t - \lambda_t s_t$ with $\lambda_t = (g_t + s_t^\top(y_t - x_t) + \rho)/\|s_t\|^2$, which matches the algorithm.

Next, $H_t \supseteq \mathcal{X}_\rho$: for any $x \in \mathcal{X}_\rho$, by convexity of $g$ and $s_t \in \partial g(x_t)$, $g_t + s_t^\top(x - x_t) + \rho \leq g(x) + \rho \leq 0$, so $x \in H_t$.

For the enhanced non-expansiveness, let $p_t = \Pi_{H_t}(y_t)$. By the variational characterization of projection onto $H_t$, for all $v \in H_t$: $(y_t - p_t)^\top(v - p_t) \leq 0$. Taking $v = x \in H_t$ and expanding $\|y_t - x\|^2 = \|y_t - p_t\|^2 + 2(y_t - p_t)^\top(p_t - x) + \|p_t - x\|^2$, we obtain $(y_t - p_t)^\top(p_t - x) \geq 0$, hence $\|y_t - x\|^2 \geq \delta_t + \|p_t - x\|^2$.

Rearranging: $\|p_t - x\|^2 \leq \|y_t - x\|^2 - \delta_t$. Finally, since $x \in \mathcal{X}_\rho \subseteq R\mathcal{B}$, non-expansiveness of $\Pi_{R\mathcal{B}}$ gives $\|x_{t+1} - x\|^2 = \|\Pi_{R\mathcal{B}}(p_t) - x\|^2 \leq \|p_t - x\|^2 \leq \|y_t - x\|^2 - \delta_t$. □

To clarify the relation of this lemma to the generalized Pythagorean projection analysis, we note that the inequality $\|p_t - x\|^2 \leq \|y_t - x\|^2 - \delta_t$ used above is the *strong* (rather than weak) form of the Pythagorean inequality for projection onto a closed convex set, which is a textbook result in convex analysis (Bauschke & Combettes, 2017, Corollary 4.10): for every closed convex set $C$ and $y \in \mathbb{R}^d$, $\|\Pi_C(y) - x\|^2 + \|y - \Pi_C(y)\|^2 \leq \|y - x\|^2$ for all $x \in C$. The same inequality is also stated as the *Generalized Pythagorean Theorem* in standard first-order-methods textbooks (e.g., Beck, 2017, Theorem 9.8; this is the form often cited in OCO under the name "generalized Pythagorean projection"). Prior worst-case OCO analyses of the Polyak feasibility step use only the weak form $\|p_t - x\|^2 \leq \|y_t - x\|^2$ because the additional slack $\delta_t$ is not needed to derive the $O(\sqrt{T})$ rate. Our contribution at this point is therefore not to discover a new inequality, but to recognise that the slack $\delta_t = \|y_t - \Pi_{H_t}(y_t)\|^2$ equals a measurable, algorithmically meaningful Polyak-projection quantity that, once summed, gives a strictly negative correction $-\frac{1}{2\eta}\mathcal{P}_T$ in the regret bound. To our knowledge this is the first OCO regret bound to make the strong-inequality slack explicit as a named, data-dependent quantity that can be evaluated post-hoc on any trajectory.

### A.2  Lemma 2: Constraint Error Bound

**Lemma 2** (Constraint error bound). *Let Assumptions 3–4 hold and $\rho \in [0, \epsilon]$. Then for all $x \in R\mathcal{B}$,*

$$\text{dist}(x, \mathcal{X}_\rho) \leq \frac{1}{\sigma}[g(x) + \rho]_+. \tag{17}$$

*Proof.* If $g(x) + \rho \leq 0$, then $x \in \mathcal{X}_\rho$ and both sides are zero. Assume $g_\rho(x) := g(x) + \rho > 0$, i.e., $x \notin \mathcal{X}_\rho$. Let $v = \Pi_{\mathcal{X}_\rho}(x)$.

Suppose $g(v) + \rho < 0$. Since $g$ is $G_g$-Lipschitz (Assumption 3), let $\delta_0 = -(g(v) + \rho)/G_g > 0$. Then for all $w \in v + \delta_0 \mathcal{B}$, $g(w) + \rho \leq g(v) + \rho + G_g \delta_0 = 0$, so $v + \delta_0 \mathcal{B} \subseteq \mathcal{X}_\rho$. Taking $v' = v + \delta_0(x - v)/\|x - v\| \in \mathcal{X}_\rho$ gives $\|v' - x\| = \|x - v\| - \delta_0 < \text{dist}(x, \mathcal{X}_\rho)$, contradicting the definition of $v$. Hence $g(v) = -\rho$.

The Slater condition for $\mathcal{X}_\rho$ holds: by Assumption 4, $\{x : g(x) = -\epsilon\} \neq \emptyset$; if $\min_x g(x) = -\epsilon$, then there exists $\hat{x}$ with $0 \in \partial g(\hat{x})$ and $g(\hat{x}) = -\epsilon$, but Assumption 4 requires $\|s\| \geq \sigma > 0$ for all $s \in \partial g(\hat{x})$, a contradiction.

Hence $\min_x g(x) < -\epsilon \leq -\rho$, and Slater's condition holds. By Rockafellar (1970) (Theorem 23.7), there exist $\tilde{\mu} > 0$ and $\tilde{s}_v \in \partial g(v)$ such that

$$x - v = \tilde{\mu}\,\tilde{s}_v\,. \tag{18}$$

When $\rho = \epsilon$: $g(v) = -\epsilon$, so $v \in \mathcal{X}'$ and $\|\tilde{s}_v\| \geq \sigma$ directly by Assumption 4.

When $\rho < \epsilon$: let $C_\epsilon = \{u : g(u) \leq -\epsilon\}$. Since $g(v) = -\rho > -\epsilon$, we have $v \notin C_\epsilon$. Let $w = \Pi_{C_\epsilon}(v)$. By the same boundary argument as Step 1, $g(w) = -\epsilon$. By the normal cone characterization (Slater's condition for $C_\epsilon$ holds since $\min g < -\epsilon$), there exist $\mu > 0$ and $\hat{s}_w \in \partial g(w)$ with $v - w = \mu\,\hat{s}_w$. Since $g(w) = -\epsilon$, Assumption 4 gives $\|\hat{s}_w\| \geq \sigma$.

By monotonicity of $\partial g$ (Rockafellar, 1970): $(\tilde{s}_v - \hat{s}_w)^\top(v - w) \geq 0$. Substituting $v - w = \mu\hat{s}_w$ and dividing by $\mu > 0$: $\tilde{s}_v^\top \hat{s}_w \geq \|\hat{s}_w\|^2$. By Cauchy–Schwarz: $\|\tilde{s}_v\|\|\hat{s}_w\| \geq \tilde{s}_v^\top \hat{s}_w \geq \|\hat{s}_w\|^2$, so $\|\tilde{s}_v\| \geq \|\hat{s}_w\| \geq \sigma$.

By convexity of $g$ and $g(v) + \rho = 0$: $g_\rho(x) = g(x) + \rho \geq g(v) + \tilde{s}_v^\top(x - v) + \rho = \tilde{s}_v^\top(\tilde{\mu}\,\tilde{s}_v) = \tilde{\mu}\|\tilde{s}_v\|^2$. Hence $\tilde{\mu} \leq g_\rho(x)/\|\tilde{s}_v\|^2$, and by equation 18: $\mathrm{dist}(x, \mathcal{X}_\rho) = \|x - v\| = \tilde{\mu}\|\tilde{s}_v\| \leq \frac{g_\rho(x)}{\|\tilde{s}_v\|} \leq \frac{g_\rho(x)}{\sigma} = \frac{[g(x)+\rho]_+}{\sigma}$. $\qquad\square$

## A.3  Lemma 3: Geometric Contraction of the Polyak Step

**Lemma 3** (Geometric contraction). *Let Assumptions 1–4 hold, $\rho \in [0, \epsilon]$, $x \in R\mathcal{B}$, and $s \in \partial g(x)$ with $s \neq 0$. Let $x^+ = \Pi_{R\mathcal{B}}(x - [g(x) + \rho]_+ s/\|s\|^2)$. Then*

$$\mathrm{dist}^2(x^+, \mathcal{X}_\rho) \leq \gamma \cdot \mathrm{dist}^2(x, \mathcal{X}_\rho)\,. \tag{19}$$

*Proof.* If $g(x) + \rho \leq 0$, then $x^+ = x \in \mathcal{X}_\rho$ and both sides are zero. If $g(x) + \rho > 0$, let $v = \Pi_{\mathcal{X}_\rho}(x)$ and $\tilde{x} = x - \frac{g(x)+\rho}{\|s\|^2}s$. By non-expansiveness of $\Pi_{R\mathcal{B}}$ (since $v \in R\mathcal{B}$): $\mathrm{dist}^2(x^+, \mathcal{X}_\rho) \leq \|\tilde{x} - v\|^2$.

Expanding $\|\tilde{x} - v\|^2$ and using $s^\top(x - v) \geq g(x) - g(v) \geq g(x) + \rho$:

$$\|\tilde{x} - v\|^2 \leq \|x - v\|^2 - \frac{(g(x) + \rho)^2}{\|s\|^2} \leq \mathrm{dist}^2(x, \mathcal{X}_\rho) - \frac{(g(x) + \rho)^2}{G_g^2}\,.$$

By Lemma 2, $(g(x) + \rho)^2 \geq \sigma^2\,\mathrm{dist}^2(x, \mathcal{X}_\rho)$, so the last expression is $\leq (1 - \sigma^2/G_g^2)\,\mathrm{dist}^2(x, \mathcal{X}_\rho) = \gamma\,\mathrm{dist}^2(x, \mathcal{X}_\rho)$. $\qquad\square$

## A.4  Lemma 4: Feasibility Recursion

**Lemma 4** (Feasibility recursion). *Under Assumptions 1–4 with $\rho \in [0, \epsilon]$, Algorithm 1 satisfies for all $t \geq 1$:*

$$\mathrm{dist}(x_{t+1}, \mathcal{X}_\rho) \leq \sqrt{\gamma}\,\mathrm{dist}(x_t, \mathcal{X}_\rho) + \eta\|\nabla f_t(x_t)\|\,. \tag{20}$$

*Proof.* Define the virtual iterate $z_{t+1} := \Pi_{R\mathcal{B}}(x_t - [g_t + \rho]_+ s_t/\|s_t\|^2)$ when $s_t \neq 0$, and $z_{t+1} := x_t$ when $s_t = 0$. By the triangle inequality: $\mathrm{dist}(x_{t+1}, \mathcal{X}_\rho) \leq \|x_{t+1} - z_{t+1}\| + \mathrm{dist}(z_{t+1}, \mathcal{X}_\rho)$.

By Lemma 3 (when $s_t \neq 0$) or by $z_{t+1} = x_t \in \mathcal{X}_\rho$ (when $s_t = 0$), we get $\mathrm{dist}(z_{t+1}, \mathcal{X}_\rho) \leq \sqrt{\gamma}\,\mathrm{dist}(x_t, \mathcal{X}_\rho)$.

In both cases ($s_t = 0$ and $s_t \neq 0$), one can write $x_{t+1} = \Pi_{R\mathcal{B}}(\Pi_{H_t}(y_t))$ and $z_{t+1} = \Pi_{R\mathcal{B}}(\Pi_{H_t}(x_t))$. By non-expansiveness of $\Pi_{R\mathcal{B}} \circ \Pi_{H_t}$: $\|x_{t+1} - z_{t+1}\| \leq \|y_t - x_t\| = \eta\|\nabla f_t(x_t)\|$.

Combining yields equation 20. $\qquad\square$

## A.5  Lemma 5: Recursion Expansion

**Lemma 5** (Recursion expansion). *Under the conditions of Lemma 4, for all $t \geq 1$:*

$$\mathrm{dist}(x_{t+1}, \mathcal{X}_\rho) \leq \gamma^{t/2}\,\mathrm{dist}(x_1, \mathcal{X}_\rho) + \frac{\eta G_f}{\xi}\,, \tag{21}$$

*where $\xi = 1 - \sqrt{\gamma} > 0$.*

*Proof.* Let $d_t = \text{dist}(x_t, \mathcal{X}_\rho)$. By Lemma 4 and Assumption 2: $d_{t+1} \le \sqrt{\gamma}\,d_t + \eta G_f$. Unrolling this linear recursion by induction: $d_{t+1} \le (\sqrt{\gamma})^t d_1 + \eta G_f \sum_{k=0}^{t-1}(\sqrt{\gamma})^k \le \gamma^{t/2}d_1 + \eta G_f/\xi$, where the geometric series is bounded by $1/(1 - \sqrt{\gamma}) = 1/\xi$. $\qquad\square$

## A.6 Proof of Theorem 1

*Proof of Theorem 1.* Decompose the regret as

$$\text{Reg}_T = \underbrace{\sum_{t=1}^{T}(f_t(x_t) - f_t(x_\rho^\star))}_{\text{Term I}} + \underbrace{\sum_{t=1}^{T}(f_t(x_\rho^\star) - f_t(x^\star))}_{\text{Term II}}, \tag{22}$$

where $x_\rho^\star \in \arg\min_{x \in \mathcal{X}_\rho} \sum_t f_t(x)$.

By convexity of $f_t$: $f_t(x_t) - f_t(x_\rho^\star) \le \nabla f_t(x_t)^\top (x_t - x_\rho^\star)$. By Lemma 1 with $x = x_\rho^\star \in \mathcal{X}_\rho$: $\|x_{t+1} - x_\rho^\star\|^2 \le \|y_t - x_\rho^\star\|^2 - \delta_t$. Expanding $\|y_t - x_\rho^\star\|^2$ with $y_t = x_t - \eta\nabla f_t(x_t)$:

$$\|y_t - x_\rho^\star\|^2 = \|x_t - x_\rho^\star\|^2 - 2\eta\,\nabla f_t(x_t)^\top (x_t - x_\rho^\star) + \eta^2\|\nabla f_t(x_t)\|^2.$$

Combining and rearranging:

$$f_t(x_t) - f_t(x_\rho^\star) \le \frac{1}{2\eta}\big(\|x_t - x_\rho^\star\|^2 - \|x_{t+1} - x_\rho^\star\|^2\big) + \frac{\eta}{2}\|\nabla f_t(x_t)\|^2 - \frac{\delta_t}{2\eta}.$$

Summing over $t = 1, \ldots, T$ (telescoping) and using $\|x_1 - x_\rho^\star\| \le 2R$, $\|x_{T+1} - x_\rho^\star\|^2 \ge 0$:

$$\text{Term I} \le \frac{2R^2}{\eta} + \frac{\eta}{2}\mathcal{G}_T - \frac{1}{2\eta}\mathcal{P}_T. \tag{23}$$

The prior proof applies the weak (rather than strong) Pythagorean inequality $\|x_{t+1} - x_\rho^\star\|^2 \le \|y_t - x_\rho^\star\|^2$ (which does not retain $\delta_t$) and bounds $\|\nabla f_t(x_t)\|^2 \le G_f^2$ (which collapses the data-dependent sum to $G_f^2 T$), yielding Term I $\le \frac{2R^2}{\eta} + \frac{\eta}{2}G_f^2 T$. Our proof retains both quantities.

By definition of $x_\rho^\star$: $\sum_t f_t(x_\rho^\star) \le \sum_t f_t(\Pi_{\mathcal{X}_\rho}(x^\star))$. By $G_f$-Lipschitz continuity of each $f_t$: $f_t(\Pi_{\mathcal{X}_\rho}(x^\star)) - f_t(x^\star) \le G_f\,\text{dist}(x^\star, \mathcal{X}_\rho)$. By Lemma 2 with $g(x^\star) \le 0$: $\text{dist}(x^\star, \mathcal{X}_\rho) \le \rho/\sigma$. Hence Term II $\le G_f\rho T/\sigma$. Combining with equation 23 yields equation 6.

By Lemma 5 (replacing $t$ by $t-1$), for all $t \ge 1$: $\text{dist}(x_t, \mathcal{X}_\rho) \le \gamma^{(t-1)/2}\,\text{dist}(x_1, \mathcal{X}_\rho) + \eta G_f/\xi$. Since $g$ is $G_g$-Lipschitz and $g(\Pi_{\mathcal{X}_\rho}(x_t)) \le -\rho$: $g(x_t) \le G_g\,\text{dist}(x_t, \mathcal{X}_\rho) - \rho$, which gives equation 7. $\qquad\square$

## A.7 Proofs of Corollaries

*Proof of Corollary 1.* Feasibility: $g(x_1) \le -\alpha \le -\rho$ gives $\text{dist}(x_1, \mathcal{X}_\rho) = 0$. Substituting into equation 7: $g(x_t) \le \eta G_g G_f/\xi - \rho$. With $\eta = \xi\rho/(G_f G_g)$: $\eta G_g G_f/\xi = \rho$, so $g(x_t) \le 0$.

*Regret:* $\frac{2R^2}{\eta} = \frac{2R^2 G_f G_g \sqrt{T}}{\xi\alpha}$, $\quad \frac{\eta}{2}\mathcal{G}_T = \frac{\xi\alpha}{2G_f G_g \sqrt{T}}\mathcal{G}_T$, $\quad \frac{1}{2\eta}\mathcal{P}_T = \frac{G_f G_g \sqrt{T}}{2\xi\alpha}\mathcal{P}_T$, $\quad \frac{G_f\rho}{\sigma}T = \frac{G_f\alpha}{\sigma}\sqrt{T}$. Combining yields equation 8. $\qquad\square$

*Proof of Corollary 2.* The feasibility analysis is identical to that of prior work (since equation 7 is unchanged). The regret part follows by substituting $\eta = \xi\epsilon/(2G_f G_g \sqrt{T})$, $\rho = \epsilon/\sqrt{T}$ into equation 6 and using $\mathcal{G}_T \le G_f^2 T$. $\qquad\square$

*Proof of Corollary 3.* With $\rho = 0$ and $\eta = 2R/(G_f\sqrt{T})$: $\frac{2R^2}{\eta} = RG_f\sqrt{T}$, $\quad \frac{\eta}{2}\mathcal{G}_T = \frac{R\mathcal{G}_T}{G_f\sqrt{T}}$, $\quad \frac{1}{2\eta}\mathcal{P}_T = \frac{G_f\sqrt{T}}{4R}\mathcal{P}_T$. The feasibility bound follows from equation 7 with $\rho = 0$. $\qquad\square$

*Proof of Corollary 4.* Direct subtraction: $\left(\frac{\eta}{2}G_f^2 T\right) - \left(\frac{\eta}{2}\mathcal{G}_T - \frac{1}{2\eta}\mathcal{P}_T\right) = \frac{\eta}{2}(G_f^2 T - \mathcal{G}_T) + \frac{1}{2\eta}\mathcal{P}_T$. Both terms are non-negative since $\mathcal{G}_T \le G_f^2 T$ and $\mathcal{P}_T \ge 0$. $\qquad\square$

*Proof of Proposition 1.* When $t \notin \mathcal{A}$, $\delta_t = 0$. When $t \in \mathcal{A}$, $\delta_t = (g_t + s_t^\top(y_t - x_t) + \rho)^2/\|s_t\|^2 \ge (g_t + s_t^\top(y_t - x_t) + \rho)^2/G_g^2$ by Assumption 3. $\qquad\square$

*Proof of Proposition 2.* (a) follows from $\delta_t = 0$ for all $t$. (b) follows from Proposition 1 with $|\mathcal{A}| = T$. $\qquad\square$

## A.8   Proof of Theorem 2 (AdaOGD-PFS)

*Proof of Theorem 2.* Lemmas 1–3 hold unchanged, since they do not depend on the step size $\eta$. The proof adapts the telescoping argument of Theorem 1 to the time-varying step size $\eta_t = c/\sqrt{S_t}$.

Decompose $\text{Reg}_T$ as in equation 22.

By convexity and Lemma 1 (which holds for any $\eta_t$):

$$f_t(x_t) - f_t(x_\rho^\star) \le \frac{1}{2\eta_t}\left(\|x_t - x_\rho^\star\|^2 - \|x_{t+1} - x_\rho^\star\|^2\right) + \frac{\eta_t}{2}\|\nabla f_t(x_t)\|^2 - \frac{\delta_t}{2\eta_t}.$$

Summing over $t = 1, \dots, T$:

$$\text{Term I} \le \underbrace{\sum_{t=1}^T \frac{1}{2\eta_t}\left(\|x_t - x_\rho^\star\|^2 - \|x_{t+1} - x_\rho^\star\|^2\right)}_{(A)} + \underbrace{\sum_{t=1}^T \frac{\eta_t}{2}\|\nabla f_t(x_t)\|^2}_{(B)} - \sum_{t=1}^T \frac{\delta_t}{2\eta_t}. \tag{24}$$

Let $a_t := 1/(2\eta_t) = \sqrt{S_t}/(2c)$, which is non-decreasing since $S_t$ is non-decreasing. Let $D_t := \|x_t - x_\rho^\star\|^2 \le 4R^2$. By Abel's summation formula:

$$(A) = a_1 D_1 - a_T D_{T+1} + \sum_{t=2}^T D_t(a_t - a_{t-1}) \le 4R^2\left(a_1 + \sum_{t=2}^T(a_t - a_{t-1})\right) = 4R^2\, a_T = \frac{2R^2}{c}\sqrt{S_{T+1}}.$$

Since $S_{T+1} = \epsilon_0 + \mathcal{G}_T$, we get $(A) \le \frac{2R^2}{c}\sqrt{\mathcal{G}_T + \epsilon_0}$.

For non-negative $b_t := \|\nabla f_t(x_t)\|^2$ and $S_t = \epsilon_0 + \sum_{i<t} b_i$:

$$\sum_{t=1}^T \frac{b_t}{\sqrt{S_t}} \le 2\left(\sqrt{S_{T+1}} - \sqrt{S_1}\right) = 2\left(\sqrt{\mathcal{G}_T + \epsilon_0} - \sqrt{\epsilon_0}\right).$$

Therefore $(B) = \frac{c}{2}\sum_t b_t/\sqrt{S_t} \le c\left(\sqrt{\mathcal{G}_T + \epsilon_0} - \sqrt{\epsilon_0}\right)$.

We note in passing that the inequality $\sum_{t=1}^T b_t/\sqrt{S_t} \le 2(\sqrt{S_{T+1}} - \sqrt{S_1})$ used above is the standard telescoping bound that holds when the denominator includes the current $b_t$, i.e., $S_{t+1}$ instead of $S_t$ (Duchi et al., 2011, Lemma 4). With the past-gradient denominator $S_t$ used by Algorithm 2, the same telescoping yields the slightly weaker bound

$$\sum_{t=1}^T \frac{b_t}{\sqrt{S_t}} = \sum_{t=1}^T \frac{b_t}{\sqrt{S_{t+1}}} + \sum_{t=1}^T b_t\left(\frac{1}{\sqrt{S_t}} - \frac{1}{\sqrt{S_{t+1}}}\right) \le 2\left(\sqrt{S_{T+1}} - \sqrt{S_1}\right) + \frac{G_f^2}{\sqrt{\epsilon_0}},$$

because $b_t \le G_f^2$ and $\sum_t(1/\sqrt{S_t} - 1/\sqrt{S_{t+1}}) = 1/\sqrt{S_1} - 1/\sqrt{S_{T+1}} \le 1/\sqrt{\epsilon_0}$ telescopes. Substituting back gives $(B) \le c\left(\sqrt{\mathcal{G}_T + \epsilon_0} - \sqrt{\epsilon_0}\right) + cG_f^2/(2\sqrt{\epsilon_0})$, which produces the $cG_f^2/(2\sqrt{\epsilon_0})$ overhead now made explicit in equation 13.

$$\text{Term I} \le \frac{2R^2}{c}\sqrt{\mathcal{G}_T + \epsilon_0} + c\sqrt{\mathcal{G}_T + \epsilon_0} - c\sqrt{\epsilon_0} + \frac{cG_f^2}{2\sqrt{\epsilon_0}} - \sum_{t=1}^T \frac{\delta_t}{2\eta_t} \le \left(\frac{2R^2}{c} + c\right)\sqrt{\mathcal{G}_T + \epsilon_0} + \frac{cG_f^2}{2\sqrt{\epsilon_0}} - \sum_{t=1}^T \frac{\delta_t}{2\eta_t}.$$

Identical to the proof of Theorem 1: Term II $\leq G_f \rho T / \sigma$.

Combining yields equation 13.

Since $\eta_t$ is non-increasing, $\eta_1 = c/\sqrt{\epsilon_0} = \max_t \eta_t$. The feasibility recursion (Lemma 4) gives $\text{dist}(x_{t+1}, \mathcal{X}_\rho) \leq \sqrt{\gamma}\,\text{dist}(x_t, \mathcal{X}_\rho) + \eta_t \|\nabla f_t(x_t)\| \leq \sqrt{\gamma}\,\text{dist}(x_t, \mathcal{X}_\rho) + \eta_1 G_f$, which, by the same expansion as Lemma 5, yields $\text{dist}(x_t, \mathcal{X}_\rho) \leq \gamma^{(t-1)/2}\,\text{dist}(x_1, \mathcal{X}_\rho) + \eta_1 G_f/\xi$. Applying the Lipschitz bound $g(x_t) \leq G_g\,\text{dist}(x_t, \mathcal{X}_\rho) - \rho$ gives equation 14. $\qquad\square$

*Proof of Corollary 5. Feasibility:* $g(x_1) \leq -\alpha \leq -\rho$ gives $\text{dist}(x_1, \mathcal{X}_\rho) = 0$. By equation 14: $g(x_t) \leq c\,G_g G_f/(\xi\sqrt{\epsilon_0}) - \rho$. With $c = R\sqrt{2}$ and $\epsilon_0 = 2G_g^2 G_f^2 R^2/(\xi^2 \rho^2)$: $c\,G_g G_f/(\xi\sqrt{\epsilon_0}) = R\sqrt{2} \cdot G_g G_f/(\xi \cdot G_g G_f R\sqrt{2}/(\xi\rho)) = \rho$. Hence $g(x_t) \leq 0$.

Substitute $c = R\sqrt{2}$ into equation 13: $\frac{2R^2}{R\sqrt{2}} + R\sqrt{2} = 2R\sqrt{2}$. The past-gradient overhead $cG_f^2/(2\sqrt{\epsilon_0})$ evaluates, under the chosen $\epsilon_0 = 2G_g^2 G_f^2 R^2/(\xi^2 \rho^2)$ and $c = R\sqrt{2}$, to $\xi\rho G_f/(2G_g)$; substituting $\rho = \alpha/\sqrt{T}$ gives $\xi\alpha G_f/(2G_g\sqrt{T})$, which is absorbed into the leading $\sqrt{T}$ term in equation 15 and contributes a vanishing additive constant relative to the dominant $2R\sqrt{2\mathcal{G}_T + 2\epsilon_0}$ piece. $\qquad\square$

# B   Additional Experiments

## B.1   AdaOGD-PFS vs Fixed-Step-Size OGD-PFS

Table 3 shows that AdaOGD-PFS achieves bounds comparable to Theorem 1 while using an adaptive step size that does not require knowledge of $G_f$. On the halfspace problem, AdaOGD-PFS achieves *lower actual regret* (148 vs. 173 for OGD-PFS; both measured on the same problem instance) than fixed-step-size OGD-PFS, demonstrating the benefit of adapting $\eta_t$ to the observed gradient magnitudes. On the ball problems, fixed-step-size OGD-PFS has lower regret (235 vs. 373 for AdaOGD-PFS), which is expected since the fixed step size $\eta = 2R/(G_f\sqrt{T})$ is tuned with knowledge of $G_f$ while AdaOGD-PFS must learn it online. The key advantage of AdaOGD-PFS is that it does not require $G_f$ as input and achieves a bound that scales with $\sqrt{\mathcal{G}_T}$ rather than $G_f\sqrt{T}$, a significant practical benefit when $G_f$ is unknown or overly conservative. Figure 2 provides a visual three-way comparison across all problem instances.

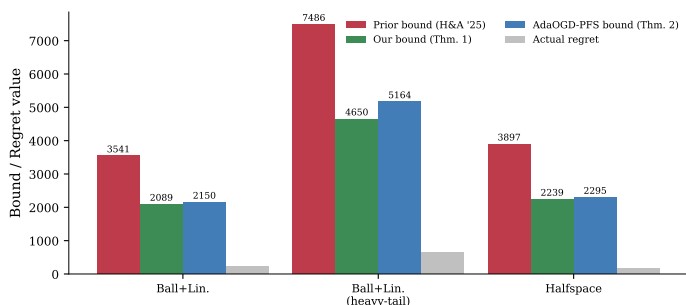

Figure 2: Three-way comparison of bounds across problem instances. Red: prior bound; green: our refined bound (Theorem 1); blue: AdaOGD-PFS bound (Theorem 2); gray: actual regret. The actual regret shown in grey is that of OGD-PFS (Algorithm 1); the actual regret of AdaOGD-PFS (Algorithm 2) at $T = 10{,}000$ is 373 (Ball+Linear), 1,173 (Ball+Linear HT), and 148 (Halfspace), as also noted in the caption of Table 3.

## B.2   Bound Component Decomposition

We further visualize the internal structure of the bound improvement from three complementary perspectives. Figure 3 shows the percentage contribution of gradient refinement versus Polyak correction for each problem. Figure 4 traces the cumulative growth of both $\mathcal{G}_t$ and $\mathcal{P}_t$ over time, revealing that the gap between $\mathcal{G}_t$ and $G_f^2 t$

widens steadily while $\mathcal{P}_t$ accumulates at a near-constant rate once the constraint becomes active. Figure 5 decomposes each bound into its additive components, making the source of tightening visually explicit.

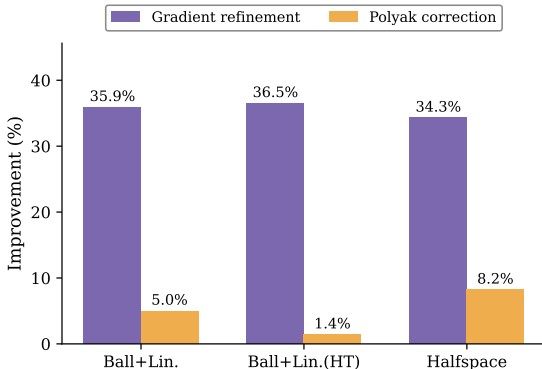

Figure 3: Decomposition of the bound improvement into gradient refinement (purple) and Polyak correction (orange) for each problem instance.

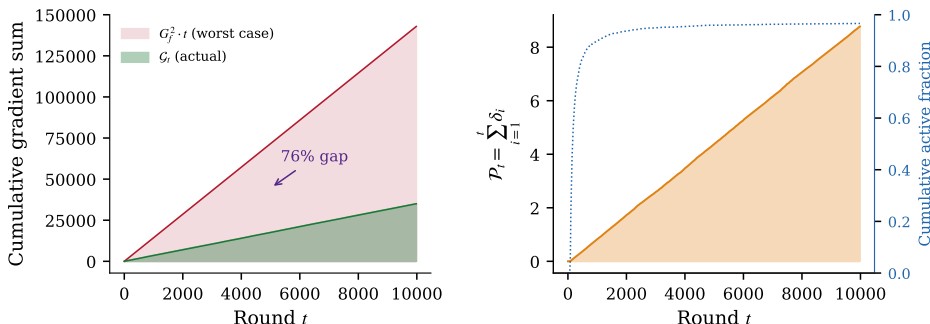

Figure 4: Left: cumulative gradient accumulation $\mathcal{G}_t$ (green) vs. worst-case $G_f^2 \cdot t$ (red), showing an $\approx 72\%$ gap. The gap percentage equals $1 - \mathcal{G}_T/(G_f^2 T)$, consistent with the ratio $\mathcal{G}_T/(G_f^2 T) \approx 0.28$ for the Ball+Linear instance in Table 3; the small numerical fluctuations across panels and seeds are within the standard deviations of Table 4. Right: cumulative Polyak correction $\mathcal{P}_t$ growing steadily over time, with the active constraint fraction (dashed) converging to $\approx 0.97$.

### B.3 Sensitivity Analysis

We examine how the bound improvement varies with the step size $\eta$ and the shrinkage parameter $\rho$. Figure 6 shows that larger $\eta$ increases both sources of improvement (gradient refinement and Polyak correction), since a larger step size amplifies the gap between observed and worst-case gradient norms and also causes more frequent constraint violations that the Polyak step must correct. Increasing $\rho$ primarily boosts the Polyak correction component, as the tighter constraint forces the feasibility projection to be active more often.

### B.4 Distributional Analysis

Finally, we inspect the per-round distributions underlying the aggregate quantities $\mathcal{G}_T$ and $\mathcal{P}_T$. Figure 7 confirms that the gradient norms $\|\nabla f_t(x_t)\|$ are concentrated well below the worst-case bound $G_f$, with the bulk of the mass near the mean rather than the tail. This concentration explains the large gradient refinement gain (34–37%). The right panel shows that the nonzero Polyak corrections $\delta_t$ span several orders of magnitude, with occasional large corrections corresponding to rounds where the gradient step pushes the iterate far outside the linearized constraint.

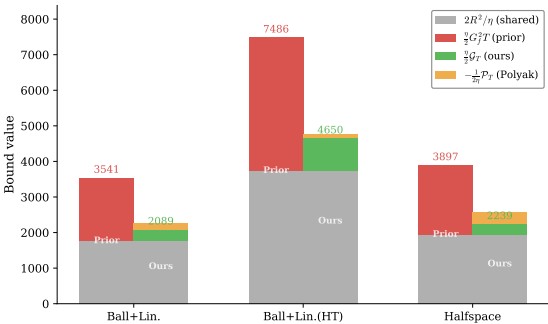

Figure 5: Bound component decomposition. Each problem shows two bars: left (prior bound) and right (our bound). The shared initial-distance term $2R^2/\eta$ (gray) is identical; the gradient term shrinks from $\frac{\eta}{2}G_f^2 T$ (red) to $\frac{\eta}{2}\mathcal{G}_T$ (green); the Polyak correction $-\frac{1}{2\eta}\mathcal{P}_T$ (orange) provides additional reduction.

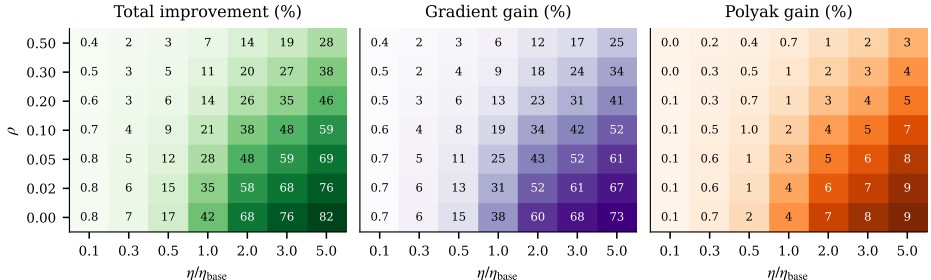

Figure 6: Sensitivity of bound improvement to step size $\eta$ and shrinkage $\rho$. Left: total improvement (%); center: gradient refinement gain; right: Polyak correction gain.

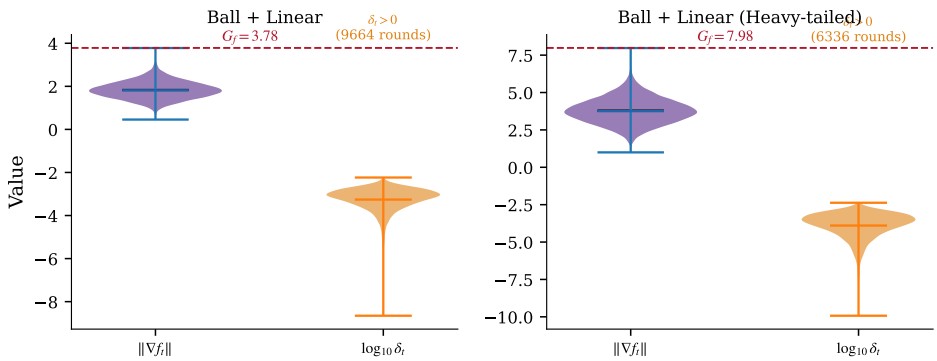

Figure 7: Distribution of per-round gradient norms $\|\nabla f_t(x_t)\|$ (left violin) compared to the worst-case bound $G_f$ (dashed red line), and log-scale distribution of nonzero Polyak corrections $\delta_t$ (right violin).

