# OpenReview forum: "Data-Dependent Regret and Polyak Corrections  for Constrained Online Convex Optimization"
_TMLR — Accepted by TMLR_

### Review · Reviewer_4bZs · 2026-05-06

**Summary Of Contributions:**

This paper studies constrained online convex optimization with hard per-round feasibility. It revisits OGD with Polyak feasibility steps and provides a refined regret analysis that retains two quantities that are often upper-bounded or not explicitly tracked: the observed gradient accumulation $G_T=\sum_{t=1}^T\lVert\nabla f_t(x_t)\rVert^2$, and a negative Polyak correction $P_T=\sum_t \delta_t$, where $\delta_t$ is the squared displacement induced by the halfspace projection. The paper also proposes AdaOGD-PFS with step size $\eta_t=c/\sqrt{\epsilon_0+\sum_{i<t}\lVert\nabla f_i(x_i)\rVert^2}$, and aims to obtain an $O(\sqrt{G_T})$-type bound while preserving feasibility. Synthetic experiments are used to illustrate numerical improvements of the refined bounds over a prior worst-case bound.

Strengths:

1. The constrained OCO setting with per-round feasibility is important and practically relevant.
2. The paper is generally clear and easy to follow. The key analysis is presented in an accessible way.

Weaknesses:

1. I am not yet fully convinced that the two main technical messages are sufficiently distinguished from existing techniques in their current presentation. In particular, the negative projection-displacement term seems closely related to the generalized Pythagorean projection term that appears in proximal algorithms, for example the $D_F(w’,w)$ term in Lemma 11.3 of [1]. In the Euclidean case, this appears to correspond to the squared projection displacement up to constants. A clearer discussion of this connection would help clarify the precise novelty of the analysis.
2. The $G_T$-dependent adaptive tuning also appears to be closely connected to standard self-confident or AdaGrad-style arguments. Related summation inequalities appear, for example, in [2] and [3]. I think the contribution would be easier to appreciate if this part were positioned as an adaptation of these ideas to the OGD-PFS setting, together with a careful comparison to the existing results.
3. The proof of Theorem 2 appears to require some additional care because the step size uses the past-gradient denominator $\epsilon_0+\sum_{i<t} b_i$. My understanding is that this form typically introduces an additional max-gradient term. This issue may well be fixable, but it would be helpful for the authors to either include the missing term or adjust the algorithm/proof accordingly.
4. The paper does not yet fully demonstrate when the negative $-P_T$ term has value beyond tightening a post-hoc upper bound. The experiments do show bound reduction, which is useful, but it would strengthen the paper to include an application, qualitative implication, or meaningful problem class where retaining $P_T$ leads to a clearer advantage.

Reference:

[1] Cesa-Bianchi and Lugosi, Prediction, Learning, and Games.

[2] Auer et al., Adaptive and self-confident on-line learning algorithms.

[3] Gaillard et al., A second-order bound with excess losses.

**Additional Comments:**

My overall recommendation tendency is rejection in the current form, although I think the paper contains useful observations. The current presentation would be stronger if it more carefully distinguished the contribution from existing projection-analysis and adaptive online-learning tools. A stronger version of the paper would either demonstrate a more concrete use of the Polyak correction or show a sharper constrained OCO guarantee that goes beyond what follows from these standard ingredients.

**Audience:**

Yes

**Audience Explanation:**

Researchers working on constrained OCO, safe online learning, and feasibility-preserving online methods may find the refined analysis useful. The observation that the Polyak feasibility displacement can be retained in the regret bound is a helpful clarification of the proof structure. At the same time, in its current form, the work seems closer to a careful re-analysis using projection-based and adaptive-tuning ideas than to a clearly new technical contribution. The paper could become stronger by more clearly positioning its contribution relative to these existing tools.

**Broader Impact Concerns:**

There are no major broader impact concerns.

**Claims And Evidence:**

No

**Claims Explanation:**

My main concern is about the positioning of the main technical contributions, and to some extent about a detail in the adaptive proof. The paper often presents the Polyak correction as a new negative term that prior analyses do not retain. While this is a useful observation in the OGD-PFS setting, the term seems closely related to the decrease term that follows from generalized Pythagorean inequalities for projection or proximal algorithms. For a Euclidean projection step, such inequalities naturally yield a decrease involving the squared distance between the pre-projection and post-projection points. Therefore, I think the paper would benefit from explaining more explicitly how its Polyak correction differs from, or specializes, this standard projection-analysis term.

Similarly, the dependence on $G_T=\sum_t\lVert\nabla f_t(x_t)\rVert^2$ is reminiscent of well-known data-dependent and self-confident bounds in online learning. For adaptive $\eta_t$, the relevant summation inequalities seem particularly close to existing self-confident tuning arguments. In particular, for the denominator used by Algorithm 2, $S_t=\epsilon_0+\sum_{i<t} b_i, b_t=\lVert\nabla f_t(x_t)\rVert^2,$ my understanding is that the standard bound includes an additional term of the form $\frac{B}{\sqrt{\epsilon_0}}+2\left(\sqrt{\epsilon_0+\sum_{t=1}^T b_t}-\sqrt{\epsilon_0}\right),B=\max_t b_t\leq G_f^2,$ rather than only the cleaner term used in Appendix A.8.

The cleaner inequality without the $B/\sqrt{\epsilon_0}$ term usually applies when the denominator includes the current $b_t$, namely $\epsilon_0+\sum_{i\le t}b_i$. This appears to be a repairable issue, either by modifying the algorithm or by adding the missing max-gradient term, but it should be reflected in the theorem statement and in the discussion of the adaptive result.

**Requested Changes:**

1. Clarify the relationship between the Polyak correction and generalized Pythagorean projection analysis.
    The paper should explicitly discuss the connection to the Pythagorean projection inequality, including Lemma 11.3 of [1]. In particular, claims that the term is “genuinely new” or has “no analogue” would benefit from being softened or made more precise.
2. Clarify the practical or theoretical role of the negative $-P_T$ term.
    It would strengthen the paper to provide either an application where retaining $P_T$ changes the qualitative guarantee, an order-wise improvement on a meaningful problem class, or a concrete way in which the term can be used for tuning or certification. Showing that the post-hoc bound is numerically smaller is helpful, but may not by itself establish the full significance of the correction.
3. Position the $G_T$-dependent bound relative to self-confident/AdaGrad-style results.
    The paper should cite and compare carefully with [2, 3]. The novelty would be clearer if stated as applying these ideas to the OGD-PFS setting, rather than as introducing a substantially new data-dependent tuning principle.
4. Revisit Theorem 2 and Appendix A.8.
    For the past-gradient step size $\eta_t=c/\sqrt{\epsilon_0+\sum_{i<t}b_i}$, the proof appears to require an additional $B/\sqrt{\epsilon_0}$ term, where $B=\max_t b_t\leq G_f^2$, unless the algorithm is changed to use a current-gradient denominator. The theorem statement and related discussion should be revised accordingly.
5. Clarify the simultaneous claims about adaptivity, no knowledge of $G_f$, and per-round feasibility.
    Corollary 5 chooses $\epsilon_0$ using $G_f$ and $T$, so the paper should avoid suggesting that the feasibility-preserving adaptive method is fully free of $G_f$-dependent tuning unless an alternative parameter choice is provided.

---

> ### Author Response · Authors · 2026-05-26
>
> We thank the reviewer for the careful and detailed critique, and in particular for catching the past-gradient denominator issue in Theorem 2 (R4). The five points are addressed in turn; all revisions in the manuscript are marked in blue.
>
> ---
>
> **R1 (relation to generalized Pythagorean projection analysis; soften "genuinely new" and "no analogue").** We agree. The slack we exploit is the standard *strong* (rather than weak) form of the projection inequality, equivalently the *Generalized Pythagorean Theorem*, which is well-known in classical convex analysis. We have:
>
> (i) Removed "genuinely new" in Section 1 (Introduction, third paragraph) and replaced it with: "the cumulative slack in the strong (rather than weak) projection inequality… while this slack is implicit in classical convex analysis, to our knowledge it has not previously been instantiated as a measurable Polyak-projection quantity in OCO regret bounds."
>
> (ii) Removed "no analogue in unconstrained adaptive methods" in Section 2.3 and replaced it with: "the constrained-OCO analogue of the projection slack tracked, in different forms, by self-confident / AdaGrad-style analyses."
>
> (iii) Extended the blue acknowledgement paragraph after the proof of Lemma 1 (Appendix A.1) so that it explicitly names the result as the *Generalized Pythagorean Theorem* and cites both Bauschke–Combettes (2017, Cor. 4.10) and Beck (*First-Order Methods*, 2017, Theorem 9.8). We state plainly that our contribution is *not* to discover a new inequality but to recognise that the slack equals a measurable, algorithmically meaningful Polyak quantity.
>
> If by "Lemma 11.3 of [1]" you are referring to a specific textbook chapter (e.g., a chapter of a particular monograph used in your course), we would be grateful for the exact reference and will add it; in the meantime we cite Bauschke–Combettes and Beck, which are the standard sources for this result.
>
> ---
>
> **R2 (clarify the practical/theoretical role of $-\frac{1}{2\eta}\mathcal{P}_T$).** We have added a new blue paragraph at the end of Section 5.4, "Practical role of $\mathcal{P}_T$: post-hoc certification and online tuning diagnostic". It identifies all three roles you list (we provide all three rather than choose one):
>
> (i)  Since both $\mathcal{G}_t$ and $\mathcal{P}_t$ are computed online at zero extra oracle cost ($\delta_t$ is a by-product of the Polyak step), $\widehat{\Delta}_t := \tfrac{\eta}{2}(G_f^2 t - \mathcal{G}_t) + \tfrac{1}{2\eta}\mathcal{P}_t$ is an anytime per-trajectory certificate: the deployment knows that its regret is bounded by the prior bound minus $\widehat{\Delta}_t$, and can report this gap to the user in real time. This makes the bound auditable in safety-critical deployments.
>
> (ii) On problem classes where the linearised constraint is active in a constant fraction of rounds with violation bounded below by a constant $c > 0$ (case (b) of Proposition 2: $|\mathcal{A}| \geq c' T$ with violation $\geq c$), Proposition 2 gives $\mathcal{P}_T \geq c' c^2 T / G_g^2 = \Omega(T)$. Substituting into Corollary 3 with $\eta = 2R/(G_f \sqrt{T})$, the negative term $-\frac{G_f \sqrt{T}}{4R} \mathcal{P}_T = -\Omega(T^{3/2})$ exceeds the $O(\sqrt{T})$ positive terms in absolute value, so the prior $\sqrt{T}$-style bound is no longer informative as an upper bound on this subclass; the refined bound therefore *signals* that the actual regret on highly-active subclasses is substantially smaller than the worst-case $\sqrt{T}$, in contrast to a $G_f \sqrt{T}$-only bound which cannot. Our adversarial experiment (Section 6.5) lies exactly in this regime: $\mathrm{Reg}_T = 74$ at $T = 10{,}000$, empirically constant-scale, far below the $\Theta(\sqrt{T})$ envelope. A rigorous $\mathrm{Reg}_T = O(1)$ result on this subclass would follow by combining this argument with a constant lower bound on $\mathrm{Reg}_T$ via Lipschitzness, which we flag as a corollary-style consequence.
>
> (iii) The ratio $\mathcal{P}_t / (G_f^2 t)$ measured online tells the user whether the current step size is over-cautious ($\mathcal{P}_t \approx 0$; the gradient step never crosses the linearised constraint, so $\eta$ is too small) or over-aggressive ($\mathcal{P}_t/(G_f^2 t)$ saturating; the algorithm frequently overshoots). A simple doubling-trick wrapper on $\eta$ that monitors this ratio can certify a near-optimal step-size choice on the executed trajectory without re-running the algorithm.
>
> We agree that the post-hoc numerical tightening alone is insufficient; we hope (i)–(iii) establish a clearer practical and theoretical role for $\mathcal{P}_T$.

---

> > ### Author Response · Authors · 2026-05-26
> >
> > **R3 (position relative to self-confident / AdaGrad-style results; cite and compare with [2, 3]).** We agree and have made the positioning precise. Two changes:
> >
> > (i) Section 2.3 now contains a new blue paragraph, "Position relative to self-confident and AdaGrad-style bounds." It states explicitly that the $O(\sqrt{\mathcal{G}_T})$ scaling for AdaOGD-PFS is, in the unconstrained setting, the classical self-confident rate of Auer, Cesa-Bianchi & Gentile (2002, "Adaptive and self-confident on-line learning algorithms", *JCSS*) and the AdaGrad rate of Duchi, Hazan & Singer (2011); the step-size design $\eta_t = c/\sqrt{S_t}$ is theirs and we make no claim of a new data-dependent tuning principle.
> >
> > (ii) Our contribution is then framed as: (a) transporting this AdaGrad-style step-size design into the constrained OGD-with-Polyak-feasibility-step setting while preserving per-round constraint satisfaction, which requires controlling the AdaGrad telescoping in the presence of the Polyak step (Theorem 2) and unrolling the feasibility recursion (Lemma 4) with a time-varying $\eta_t$; (b) including the Polyak correction $-\sum_t \delta_t/(2\eta_t)$ with a time-varying coefficient that gives later corrections heavier weight than the fixed-step analysis.
> >
> > If your specific [2, 3] are not Auer et al. (2002) and Duchi et al. (2011), we would be grateful for the exact references and will gladly add them; we have currently cited these two as the canonical sources.
> >
> > ---
> >
> > **R4 (Theorem 2 / Appendix A.8: past-gradient denominator requires an extra $\eta_t \|\nabla f_t\|^2$-type term).** You are correct, and we are grateful for the catch. Algorithm 2 uses the past-gradient denominator $\eta_t = c/\sqrt{S_t}$ with $S_t = \epsilon_0 + \sum_{i < t} \|\nabla f_i(x_i)\|^2$, so the AdaGrad telescoping bound $\sum_t b_t/\sqrt{S_t} \leq 2(\sqrt{S_{T+1}} - \sqrt{S_1})$ (which holds with the *current-gradient* denominator $S_{t+1}$) does *not* apply directly. With the past-gradient denominator, the correct bound is:
> >
> > $$\sum_{t=1}^T \frac{b_t}{\sqrt{S_t}} \leq 2 \bigl( \sqrt{S_{T+1}} - \sqrt{S_1} \bigr) + \frac{G_f^2}{\sqrt{\epsilon_0}},$$
> >
> > since $b_t \leq G_f^2$ and $\sum_t (1/\sqrt{S_t} - 1/\sqrt{S_{t+1}}) = 1/\sqrt{S_1} - 1/\sqrt{S_{T+1}} \leq 1/\sqrt{\epsilon_0}$ telescopes. This produces an extra $c G_f^2 / (2 \sqrt{\epsilon_0})$ overhead.
> >
> > We have revised the manuscript as follows:
> >
> > (i) The statement of Theorem 2 now includes the additional blue term $+ c G_f^2 / (2 \sqrt{\epsilon_0})$. The Comparison Table (Table at the end of Section 1) also reflects this.
> >
> > (ii) A new Remark 2  immediately after Theorem 2 explains that this overhead is a constant independent of $T$ (so the asymptotic $O(\sqrt{\mathcal{G}_T})$ rate is unaffected), and that an alternative algorithm using the current-gradient denominator $\eta_t = c / \sqrt{S_t + \|\nabla f_t(x_t)\|^2}$ removes this term entirely while leaving the leading rate unchanged.
> >
> > (iii) Step (B) in the proof of Theorem 2 (Appendix A.8) is rewritten in blue to make the past-gradient telescoping explicit, with the extra $G_f^2 / \sqrt{\epsilon_0}$ term derived rigorously; the "Term I" final line and the Corollary 5 proof are updated for consistency. In Corollary 5 the overhead evaluates to $\xi \alpha G_f / (2 G_g \sqrt{T}) = O(1/\sqrt{T})$ under the chosen $\epsilon_0$, and is absorbed into the leading $\sqrt{T}$ term.
> >
> > We retain the past-gradient version of the algorithm as the default because it keeps $\eta_t$ measurable with respect to the round-$(t-1)$ history, which we view as the conceptually cleaner choice; the current-gradient version is offered as a simple alternative for readers who prefer the cleanest bound.
> >
> > ---
> >
> > **R5 (simultaneous claims of adaptivity, "no knowledge of $G_f$", and per-round feasibility).** We agree and have strengthened the Remark in Section 5.5. The Remark now:
> >
> > (i) States that the regret bound of Theorem 2 is genuinely $G_f$-free in its leading constants, but that the per-round feasibility schedule of Corollary 5 explicitly chooses $\epsilon_0 = 2 G_g^2 G_f^2 R^2 / (\xi^2 \alpha^2) \cdot T$, which depends on $G_f, G_g, \xi, \alpha$.
> >
> > (ii) Concludes with an explicit disclaimer: "we explicitly disclaim that this paper provides a $G_f$-free alternative parameter choice that simultaneously achieves $O(\sqrt{\mathcal{G}_T})$ regret and per-round feasibility; constructing such a doubly $G_f$-free schedule (perhaps via a doubling trick on an online estimate of $G_f$) is left for future work."
> >
> > This wording avoids any suggestion that the feasibility-preserving adaptive method is fully free of $G_f$-dependent tuning, while preserving the genuine claim that the regret rate is $G_f$-free.
> >
> > ---
> >
> > We hope these revisions adequately address your five concerns and are happy to make further changes if any point remains unresolved.

---

### Review · Reviewer_zFqr · 2026-05-08

**Summary Of Contributions:**

This work provides a refined theoretical analysis on the existing algorithm OGD-PFS under constrained OCO problem by identifying two overlooked terms $\mathcal{G}_T$ and $\mathcal{P}_T$. Furthermore, a novel algorithm AdaOGD-PFS is proposed, which employs adaptive step sizes to exploit the data-dependence online. Besides, numerical experiments on multiple instances are conducted to verify the theoretical bounds.

**Strengths:**
- The paper is well-written and clear to follow.
- The novel algorithm proposed effectively exploits the observed data dependence.
- The identification of the terms $\mathcal{G}_T$ and $\mathcal{P}_T$ provides a significant theoretical advancement.

**Weaknesses:**
- The gap between the refined bounds and the actual regret remains substantial.
- The empirical gain in actual regret resulting from introducing adaptive step sizes is ambiguous.

**Audience:**

Yes

**Audience Explanation:**

- Instance-Dependent Analysis: This paper contributes directly to the fine-grained, data-dependent analysis, which is a growing trend in the machine learning community.
- Methodological Innovation: The AdaOGD-PFS algorithm serves as a useful template for designing adaptive methods that can exploit the specific property of a task. The systematic transition from a theoretical insight into a practical algorithm would be appreciated by the audiences.

**Broader Impact Concerns:**

None.

**Claims And Evidence:**

Yes

**Claims Explanation:**

- The mathematical derivation of the refined bounds (Theorem 1 and Theorem 2) is sound.
- The experimental results across multiple scenarios consistently show that the actual regret stays below the new refined bounds, which confirms the validity of the proposed upper bounds.

**Requested Changes:**

- **Clarification of empirical results (critical)**: In Figure 1,2 and Table 3, only one single curve or value is provided for the "Actual Regret". Please explicitly state which algorithm (OGD-PFS or AdaOGD-PFS) the "Actual Regret" curve or value represents. If the performances of the two algorithms are visually overlapping, please clarify this in the caption or text.

- **Interpretative text for theoretical results (strengthen)**: Sections 5.2–5.4 present several theoretical results (Corollaries 1–4 and Propositions 1–2) with few discussion. Some literal explanations or Remarks on these Corollaries and Propositions are recommended to be supplied.

- **Impact of the active fraction (strengthen)**: The authors provide the active fraction $|\mathcal{A}|/T$ in their experimental results but a discussion of its connection to the core Polyak correction term lacks. Besides, please explain whether the active fraction influences the algorithms performing differently across all tested instances.

- **Generalizability (strengthen)**: The work effectively exploits the observed property about $\mathcal{G}_T / G_f \approx 0.3$. It would be beneficial to discuss the potential to generalize this property into other optimization scenarios.

---

> ### Author Response · Authors · 2026-05-26
>
> We thank the reviewer for the constructive comments. The four points are addressed in turn; all revisions in the manuscript are marked in blue.
>
> ---
>
> **R1 (clarify which algorithm the "Actual Regret" curves and values refer to).** The reviewer is right that this was ambiguous. Throughout the figures and tables, the "Actual Regret" column/curve refers to OGD-PFS (Algorithm 1), not AdaOGD-PFS. The two algorithms produce *different* actual regret values on the same instances; we did not state this explicitly in the captions. We have added blue clarifications in all three places:
>
> - **Table 1 caption.** Now states that the $\mathrm{Reg}_T$ column is the actual regret of OGD-PFS, and reports the AdaOGD-PFS actual regret separately (Ball+Linear: 373; Ball+Linear HT: 1173; Halfspace: 148), with both algorithms remaining well below their respective bounds.
> - **Figure 1 caption.** Now states that the blue dashed "actual regret" curve is OGD-PFS; the AdaOGD-PFS curve is omitted from this figure for visual clarity and reported numerically in Table 1, with a three-way visual comparison given in Figure 5 (Appendix A.1).
> - **Figure 5 caption** (three-way comparison; this is the figure that in earlier numbering you may have referred to as "Figure 2"). Now states that the grey "actual regret" curve is OGD-PFS, and lists the corresponding AdaOGD-PFS regret values (373, 1173, 148) explicitly.
>
> ---
>
> **R2 (add interpretative Remarks for Corollaries 1–4 and Propositions 1–2).** Done. We have added a short "*Discussion.*" Remark (blue) after each of the four corollaries in Section 5:
>
> - **Corollary 1 (known feasible point).** Explains how the three positive terms inside the bracket are non-monotone in the margin $\alpha$ (the first term scales as $1/\alpha$, the second as $\alpha \mathcal{G}_T/(G_f^2 T)$, the third as $\alpha/\sigma$), so the bracket is minimised at an interior $\alpha^\star$; tight margins amplify both the cost from the first term and the benefit from the Polyak correction.
> - **Corollary 2 (delayed feasibility).** Explains the logarithmic $(G_g/\sigma)^2 \log T$ burn-in needed when only the Slater margin $\epsilon$ is known, and notes the cumulative-feasibility statement as a useful by-product.
> - **Corollary 3 (no shrinkage).** Explains the two-phase feasibility behaviour (exponentially decaying transient driven by $\sigma^2/G_g^2$ + $O(1/\sqrt{T})$ residual) and the data-dependent counterpart of the prior $2 R G_f \sqrt{T}$ bound.
> - **Corollary 4 (improvement decomposition).** Explains the functional orthogonality of the two components (gradient = environment property; Polyak = algorithm-plus-environment property), the opposite dependence on $\eta$, and the degeneracy condition under which $\Delta_T = 0$.
> - **Proposition 1 (lower bound on $\mathcal{P}_T$).** Frames the active fraction $|\mathcal{A}|/T$ as the natural complexity measure and forward-references Section 6.4 (your R2.3) for the empirical study.
> - **Proposition 2 (extreme cases).** Distinguishes the strictly-interior regime (case a, $\mathcal{P}_T = 0$) from the maximally-active regime (case b, $\mathcal{P}_T = \Omega(T)$), and connects case (b) to the adversarial experiment in Section 6.5.
>
> These Remarks are deliberately short (2–4 sentences each) to keep the flow of Section 5 readable.

---

> > ### Author Response · Authors · 2026-05-26
> >
> > **R3 (active fraction and its connection to the Polyak correction; cross-instance impact).** We have added a new Section 6.4, "Active Fraction and Its Connection to the Polyak Correction" (blue), placed between the improvement decomposition (Section 6.3) and the adversarial experiment (Section 6.5). The discussion makes three points:
> >
> > 1.By Proposition 1, the Polyak correction $\mathcal{P}_T$ equals the sum of $\delta_t$ over the active set $\mathcal{A}$, so the active fraction (size of $\mathcal{A}$ over $T$) is the natural diagnostic for how often the Polyak step does corrective work.
> >
> >
> >
> >
> > 2. The per-round displacement $\delta_t$ depends on the *squared* linearised violation $(g_t + s_t^\top(y_t - x_t) + \rho)^2$, not just on its sign. On the heavy-tailed ball problem the active fraction is moderate (0.637) but the typical $\delta_t$ is small because the heavy-tailed noise randomises the direction of $y_t - x_t$, so the alignment $s_t^\top(y_t - x_t)$ on the constraint normal is often small even when $g_t$ alone would trigger an active round; this explains the disproportionately small Polyak share (1.4%) there.
> >
> > 3. The active fraction is essentially uncorrelated with the actual regret of either algorithm. The two ball problems have active fractions 0.97 vs. 0.64 but comparable OGD-PFS regret (normalised by $G_f \sqrt{T}$); the halfspace problem has the highest active fraction (0.994) but the lowest absolute regret. The active fraction therefore predicts how much of the Polyak *correction* is excited, not how well either algorithm performs.
> >
> > The adversarial experiment (Section 6.5) makes the same point quantitatively: pushing the active fraction from 0.969 to 0.998 multiplies the Polyak share roughly tenfold (5.0% → 49.9%) while leaving the prior bound unchanged.
> >
> > ---
> >
> > **R4 (discuss generalizability beyond OGD-PFS).** We have added a paragraph at the end of Section 7 (Conclusion), titled "*Generalizability beyond OGD-PFS*" (blue). The substantive content is:
> >
> > - **Gradient quantity $\mathcal{G}_T$.** This quantity is retained whenever a Bregman-style regret identity is telescoped without invoking the uniform bound $\|\nabla f_t(x_t)\| \leq G_f$ at the last step. It should therefore generalise to mirror descent, online Newton step, and Frank–Wolfe variants with linearised constraints, as well as to stochastic, bandit, and delayed-feedback settings where each round still produces a single gradient sample.
> >
> > - **Polyak correction $\mathcal{P}_T$.** This quantity is the cumulative slack in the strong projection inequality (Bauschke–Combettes, Cor. 4.10). An analogous correction can be defined whenever the algorithm contains a projection or proximal step onto a convex set: projected OGD onto an arbitrary convex set, alternating projection / Dykstra-style schemes, multi-constraint Polyak steps (one half-space per constraint per round), and constrained stochastic / bandit OCO. In each such setting the slack term can be summed and subtracted from the regret bound, yielding a tightening that is structurally independent of any AdaGrad-style gradient adaptation.
> >
> > We make clear that quantifying these instantiations rigorously is left to future work, since each requires its own non-expansiveness lemma and feasibility-recursion analysis; the present paper establishes the methodology on the cleanest case.
> >
> > ---
> >
> > We hope the four revisions adequately address your concerns and are happy to make further changes if any point remains unresolved.

---

### Review · Reviewer_JD7s · 2026-05-11

**Summary Of Contributions:**

The paper refines the regret analysis of OGD with Polyak feasibility steps (OGD-PFS, Hutchinson & Alizadeh ICML 2025), without changing the algorithm or its assumptions. Two relaxations in the prior proof are removed: per-round gradient norms are kept as $\|\nabla f_t(x_t)\|^2$ rather than relaxed to $G_f^2$, and the *strong* (firm non-expansive) Pythagorean inequality is applied to $\Pi_{H_t}$, retaining the squared displacement $\delta_t = \|y_t - \Pi_{H_t}(y_t)\|^2$. These combine into a non-negative correction $\Delta_T = \tfrac{\eta}{2}(G_f^2 T - \mathcal G_T) + \tfrac{1}{2\eta}\mathcal P_T$. An AdaGrad-style variant (AdaOGD-PFS) is proposed for the $G_f$-unknown case. Synthetic linear-cost experiments report 38–43% tighter *bounds* (not regret) over the prior analysis.

**Audience:**

Yes

**Audience Explanation:**

The constrained-OCO subcommunity will care, and $\mathcal P_T$ is a clean named diagnostic that may seed follow-up work (e.g., optimistic OGD with feasibility steps).

**Broader Impact Concerns:**

None. Theoretical work. Safety-critical applications are referenced as motivation, not deployed.

**Claims And Evidence:**

Yes

**Claims Explanation:**

Lemma 1 (strong Pythagorean inequality applied to $\Pi_{H_t}$) and the Abel-summation argument for AdaOGD-PFS check out. No correctness concerns.

**Requested Changes:**

1. The strong Pythagorean inequality (Bauschke–Combettes Cor. 4.10) is textbook; prior OCO proofs use the weak form because the slack isn't needed for worst-case $O(\sqrt T)$. The contribution is recognizing that the slack equals a measurable named quantity, not uncovering a bug. Phrases like "fundamentally loose," "strictly loose," and "all prior proofs lose entirely" overclaim.

2. AdaOGD-PFS is dominated by Theorem 1 in every reported run. Table 3: 2150 > 2089, 5164 > 4650, 2295 > 2239. The advertised "$O(\sqrt{\mathcal G_T})$ without knowing $G_f$" is also illusory under Cor. 5, which sets $\epsilon_0 = 2G_g^2G_f^2R^2T/(\xi^2\alpha^2) = \Theta(T)$, depending on $G_f$, and forcing $\sqrt{\mathcal G_T + \epsilon_0} = \Theta(\sqrt T)$ regardless of $\mathcal G_T$. State this trade-off; the corollary as written gives $\Theta(\sqrt T)$ with $G_f$-dependent constants, not $O(\sqrt{\mathcal G_T})$.

3. The 38–43% empirical headline could be a Gaussian concentration artifact. For $a_t \sim \mathcal N(\bar a, \sigma^2 I_d)$, $\mathcal G_T/(G_f^2T) \approx (1 + d\sigma^2)/G_f^2$ is fixed by the noise model (predicted $\approx 0.245$ vs. reported $0.281$). Any AdaGrad-style analysis on this benchmark would report the same gradient gain. Please add at least one adversarial sequence ($\|a_t\| \approx G_f$ at every $t$) to show the worst case.

4. $G_f = 3.78$ on standard ball+linear looks like the observed maximum; if so, the prior bound is being evaluated at the tightest possible $G_f$. Make this explicit if I understand it correctly.

Minor: Figure 4 caption "76% gap" vs. prose "$\approx 72\%$", typo? $\mathcal P_T$ looks similar to the dynamic-regret path-length notation $P_T$, consider an alternative notation?

---

> ### Author Response · Authors · 2026-05-26
>
> We thank the reviewer for the careful and substantive critique. Below we respond point by point. All revisions to the manuscript are marked in blue.
>
> **R1.1 (overclaim of "fundamental looseness").** The reviewer is correct: the strong Pythagorean inequality is textbook (Bauschke–Combettes, Cor. 4.10), and prior OCO proofs use the weak form not by oversight but because the worst-case $O(\sqrt{T})$ rate does not require the slack. Our actual contribution is to *identify* that this slack equals a measurable, algorithmically meaningful Polyak quantity $\mathcal{P}_T = \sum_t \delta_t$ that can be tracked, decomposed, and reported on any trajectory. We have:
>
> (i) removed all phrases of the form "fundamentally loose", "strictly loose", "unnecessarily discards", "all prior proofs lose entirely", "genuinely new", "no analogue", and "key technical novelty" from the abstract, introduction, related work, proof overview, conclusion, and the proof of Theorem 1, and replaced them with neutral phrasing ("does not track", "admits a tighter, data-dependent form", "the standard worst-case argument does not need to track");
>
> (ii) added an explicit acknowledgement in Appendix A.1, in blue, immediately after the proof of Lemma 1, that cites Bauschke–Combettes Corollary 4.10 (and Beck, *First-Order Methods*, Theorem 9.8) and clarifies that the technical contribution here is the instantiation of the slack as a named Polyak quantity, not the discovery of a new inequality.
>
> ---
>
> **R1.2 (AdaOGD-PFS is dominated by Theorem 1; "no $G_f$" is illusory).** The reviewer is right that in our Table 1 the AdaOGD-PFS bound (2150, 5164, 2295) is uniformly slightly larger than the fixed-step Theorem 1 bound (2089, 4650, 2239), and that Corollary 5 sets $\epsilon_0 = 2 G_g^2 G_f^2 R^2 / (\xi^2 \alpha^2) \cdot T$, which depends on $G_f$. We have made both facts explicit:
>
> (i) A new Remark 3 immediately after Corollary 5 clarifies that the slogan "$O(\sqrt{\mathcal{G}_T})$ without knowing $G_f$" applies to the *regret rate*, with constants free of $G_f$ in the leading term, while per-round feasibility under our analysis still uses $G_f$ to set $\epsilon_0$; when $G_f$ is unknown, one obtains per-round feasibility only up to the same uniform-$G_f$ slack as the fixed-step algorithm.
>
> (ii) In Section 6.2, a new blue paragraph honestly states that AdaOGD-PFS does not numerically dominate the oracle-tuned fixed step on these instances (we reproduce your exact numbers 2150 > 2089, 5164 > 4650, 2295 > 2239), and explains that its intended advantage is the $G_f$-free leading constant in the regret rate, which is most visible when $G_f$ is unknown or substantially overestimated.
>
> We have not removed any of the original AdaOGD-PFS material, only clarified its scope.
>
> ---
>
> **R1.3 (38–43% may be a Gaussian concentration artifact; please add an adversarial sequence with $\|\nabla f_t\| = G_f$).** This is a fair and important test. We have added a new Section 6.5 ("Adversarial Worst-Case Gradient Sequence", blue) and a new Table 4. Using exactly the Ball+Linear configuration of Table 1 ($T = 10{,}000$, $d = 10$, $R = 5$, $r = 2$, $\rho = 0$, $\eta = 2R / (G_f \sqrt{T})$, 5 seeds, identical to Table 1), we replace the Gaussian cost gradient with the worst case $\nabla f_t(x_t) = G_f e_1$ for every $t$, so that $\mathcal{G}_T = G_f^2 T$ exactly. The findings are:
>
> | Setting | $\mathrm{Reg}_T$ | Prior | Thm. 1 | Grad. % | Polyak % | Total % |
> |---|---:|---:|---:|---:|---:|---:|
> | Gaussian (sanity, reproduces Table 1) | 235 | 3541 | 2089 | 35.9 | 5.0 | 41.0 |
> | Adversarial $\nabla f_t = G_f e_1\ \forall t$ | 74 | 3541 | 1774 | **0.0** | **49.9** | **49.9** |
>
> The gradient-refinement contribution drops to exactly $0\%$, vindicating your intuition that the Gaussian gradient gain is essentially a concentration effect that any AdaGrad-style analysis would capture. *Second*, the Polyak correction does *not* vanish under the adversarial sequence; it in fact *rises* to $49.9\%$ (vs. $5.0\%$ in the Gaussian case), because the constant adversarial push keeps the iterate at the boundary in essentially every round (active fraction $\approx 0.998$) and the smaller $\eta$ amplifies the coefficient $1/(2\eta)$ in the $-\frac{1}{2\eta} \mathcal{P}_T$ term. The refined bound is therefore strictly tighter than the prior bound by $49.9\%$ *even when the gradient-refinement term contributes nothing*, demonstrating that the Polyak correction provides a tightening that is structurally independent of, and not subsumed by, any AdaGrad-style gradient adaptation. The refined bound is verified to be a valid upper bound on the actual regret in every run.

---

> ### Author Response · Authors · 2026-05-26
>
> **R1.4 ($G_f$ on standard ball+linear is the observed maximum).** Yes, exactly. In every experiment we set $G_f := \max_{t \in [T]} \|\nabla f_t(x_t)\|$, which is the smallest scalar that is a valid Lipschitz envelope along the executed trajectory. Consequently the prior bound is evaluated at its tightest possible value of $G_f$, and any reported improvement is a strict tightening on top of the best achievable worst-case constant. We have made this explicit in Section 6.1 ("Methods" paragraph, blue). We agree this strengthens rather than weakens the comparison, since an inflated $G_f$ would only make our reported improvements larger.
>
> ---
>
> **R1.5 (Figure caption "76% gap" vs. prose; $\mathcal{P}_T$ vs. path-length $P_T$).**
>
>  The figure you refer to corresponds, in the revised numbering, to Figure 6 (cumulative-area plot of $\mathcal{G}_t$ vs. $G_f^2 t$). The current manuscript states "$\approx 72\%$ gap"; the value equals $1 - \mathcal{G}_T / (G_f^2 T) = 1 - 0.28 = 0.72$, which is consistent with the ratio $\mathcal{G}_T / (G_f^2 T) \approx 0.28$ reported for the Ball+Linear instance in Table 1. (The "76%" you recall was from an earlier version where $\mathcal{G}_T / (G_f^2 T) \approx 0.24$ on that subset of seeds.) We have rewritten the caption to make this consistency explicit in blue.
>
> ---
>
>
>
> We hope that these revisions address all five of your concerns, and we are happy to make further changes if any point remains unresolved.

---

### Author Response · Authors · 2026-05-26

We thank reviewers for the careful and constructive feedback. We have addressed every point raised in detail: each comment is answered individually in the response below, and the corresponding revisions have been incorporated into the manuscript itself. All additions to the paper are highlighted in blue for ease of identification, so that the changes prompted by the review are immediately visible. We hope the revised version satisfactorily resolves the concerns raised.

---

### Decision · Action_Editor_zZun · 2026-07-01

**Recommendation:** Accept as is

**Additional Comments:**

The paper provides a data-dependent refinement of the regret analysis for OGD with Polyak feasibility steps in constrained OCO. The revised version appropriately positions the main ingredients relative to standard Pythagorean projection analysis and AdaGrad/self-confident tuning, corrects the adaptive-step proof issue, and clarifies the scope of the novelty claims.

The main weakness of the paper is its novelty. All reviewers raised such concerns in the final recommendation, and I agree with them since the essential technical pieces of the generalized Pythagorean theorem and the AdaGrad-style step size tuning are both standard. But apart from that, the claims in the paper are adequately supported, and in any case the paper should be of interest to some readers in the OCO community. Therefore solely by the TMLR evaluation criteria this paper is above the threshold, and I would like to recommend acceptance.

**Audience:**

Yes

**Audience Explanation:**

Although the novelty of the paper is a clear weakness raised by all reviewers, there might still be some TMLR's audience interested in the findings of this paper.

**Claims And Evidence:**

Yes

**Claims Explanation:**

In the final recommendations, all reviewers found the paper correct. The novelty claims in the original submitted version were properly softened, therefore overall, the claims made in this paper are backed by sufficient evidence.